# Low-dose carboplatin modifies the tumor microenvironment to augment CAR T cell efficacy in human prostate cancer models

L. H. Porter[1,14], J. J. Zhu [2,3,14], N. L. Lister[1], S. G. Harrison[4], S. Keerthikumar[3,5,6], D. L. Goode[3,5,6], R. Quezada Urban[3,5,6], D. J. Byrne[7], A. Azad [3,8], I. Vela[9,10,11], M. S. Hofman[3,12,13], P. J. Neeson [2,3], P. K. Darcy [2,3], J. A. Trapani[2,3], R. A. Taylor [2,3,4,13,15] ✉ & G. P. Risbridger [1,2,3,13,15] ✉

Chimeric antigen receptor (CAR) T cells have transformed the treatment landscape for hematological malignancies. However, CAR T cells are less efficient against solid tumors, largely due to poor infiltration resulting from the immunosuppressive nature of the tumor microenvironment (TME). Here, we assessed the efficacy of Lewis Y antigen (Le^Y)-specific CAR T cells in patient-derived xenograft (PDX) models of prostate cancer. In vitro, Le^Y CAR T cells directly killed organoids derived from androgen receptor (AR)-positive or AR-null PDXs. In vivo, although Le^Y CAR T cells alone did not reduce tumor growth, a single prior dose of carboplatin reduced tumor burden. Carboplatin had a pro-inflammatory effect on the TME that facilitated early and durable CAR T cell infiltration, including an altered cancer-associated fibroblast phenotype, enhanced extracellular matrix degradation and re-oriented M1 macrophage differentiation. In a PDX less sensitive to carboplatin, CAR T cell infiltration was dampened; however, a reduction in tumor burden was still observed with increased T cell activation. These findings indicate that carboplatin improves the efficacy of CAR T cell treatment, with the extent of the response dependent on changes induced within the TME.

Immune-based therapies have transformed the treatment of many cancers, but not prostate cancer[1]. Chimeric-antigen receptor (CAR) T cell therapy, in which a patient's own T cells are genetically modified to target tumor-associated antigens, has greatly improved clinical outcomes in advanced acute lymphocytic leukemia and B cell lymphoma[2,3]. However, success in treating solid tumors has been very limited due to the immunosuppressive tumor microenvironment (TME), which prevents T cell infiltration and activation at the tumor site[4]. Herein we address the challenge of how to modify the TME to enhance CAR T cell recruitment and effector function using combination approaches.

Target antigens for CAR T cells in prostate cancer include prostate-specific membrane antigen (PSMA), prostate stem cell antigen (PSCA), KLK2, and six-membrane epithelial antigen of the prostate-1 (STEAP-1), mainly as monotherapies and with limited success thus far[5–9]. In this study, we examine the efficacy of CAR T cells directed against the Lewis Y (Le^Y) glycolipid antigen. Le^Y is an onco-fetal glycolipid antigen with minimal expression in normal tissues following birth; however, >50% of human carcinomas reactivate Le^Y biosynthesis at very high levels[10]. Studies in the 1990s demonstrated that Le^Y is prominently expressed in many high-grade prostate cancers and their metastases[11,12]. Le^Y CAR T cells demonstrated high safety and tolerability in a Phase 1a clinical trial in patients with Le^Y-positive acute myeloid leukemia[13].

In addition to ensuring antigen specificity, it is important to consider the immunosuppressive TME, which can interfere with T cell

differentiation, proliferation, and exhaustion in solid tumors[14,15]. To overcome this issue, novel approaches including the engineering of CARs or therapeutic combinations have been tested in preclinical cancer models[7,16]. Promising effects have been observed with checkpoint blockade in various cancer types and are being tested in clinical trials (i.e., ACTRN12622001542785 by our team). Checkpoint inhibitors effectively block immune checkpoints on T cells and their ligands on tumor cells (i.e. PD-1/L1) and checkpoint blockade allows CAR T cells to penetrate the TME, exert their actions, and control tumor growth[16]. Alternatively, accumulating evidence suggests that chemotherapies can increase the immunogenicity of tumor cells and dampened the immunosuppressive TME. Chemotherapies, including docetaxel, oxaliplatin, and carboplatin in prostate, lung, and ovarian cancer, have been shown to increase tumor antigen presentation, increase immune checkpoint expression and promote immune cell infiltration and activation[5,7,17–22]. This shift from an immunologically cold to warm TME may prime the tumor to immunotherapies, including CAR T cells. However, studies on pre- and post-treated ovarian cancer biopsies show different patterns of immune response to chemotherapy, suggesting that TME modulation may be tumor-specific[19]. Therefore, whilst there is a strong rationale for modifying the TME with chemotherapy, further work needs to optimize their roles in enhancing immunotherapy and identifying which tumors will respond to this treatment strategy.

In this study, we tested the combination of Le[Y] CAR T cells with either carboplatin, docetaxel, or the checkpoint inhibitor nivolumab on tumor growth using patient-derived xenografts (PDXs) of prostate cancer. Our data show that carboplatin can synergize with CAR T therapy to facilitate early and persistent CAR T cell infiltration and activation within select prostate tumors to maximize therapeutic response.

## Results

Using immunohistochemistry, we examined Le[Y] expression in 800 prostate cancer specimens from 739 patients. As cell surface expression of the target antigen is critical for CAR T cells to form an immune synapse, we quantified both membrane- and cytoplasmic Le[Y] staining in this unselected patient cohort. 12.6% of tumors had ≥10% membrane Le[Y]-positive cells (the cut-off for admission into our current Phase I clinical trial in Le[Y]-positive lung cancer); more specifically, 12.3% (87/709) and 20% (6/30) of patients with localized and metastatic disease respectively demonstrated this level of membrane staining (Fig. 1a). However, in 49 PDXs from the Monash Urology Research Alliance (MURAL) collection, which are typically derived from more aggressive cases of prostate cancer[23], 57% (28/49) had ≥10% membrane Le[Y]-positive cells, with far higher expression in many specimens, based on percent cell positivity and staining intensity (Fig. 1b, c). Le[Y]-positive specimens included AR-null tumors as well as AR-positive tumors (Fig. 1c), and its expression was independent of AR or PSMA (Fig. 1c). This suggested that anti-Le[Y] CAR T cells may provide another treatment option for patients who would not benefit from AR- or PSMA-directed therapies, or who had progressed following these therapies.

To show the on-target action of Le[Y] CAR T cells in vitro, we used organoids derived from PDXs, which maintain Le[Y] expression (Fig. 2a and Supplementary Fig. 1a, b). We selected six PDXs with different phenotypes and genomic features (Fig. 1c and Supplementary Fig. 1c)[23], including three androgen receptor (AR)-positive adenocarcinomas and three AR-null tumors that express neuroendocrine markers, with a range of Le[Y] expression (Supplementary Fig. 1a, b). Coculture with Le[Y] CAR T cells, but not control empty vector T (ev-T) cells, caused morphological destruction of organoids with pronounced propidium iodine uptake, indicating secondary necrosis (Fig. 2b–d). Importantly, tumor cell death correlated with Le[Y] expression, with strongest PI uptake in organoids with high (≥ 90%) Le[Y]

expression (Fig. 2d, Supplementary Fig. 1a, b). In contrast, PDX-435.1A-Cx organoids, with 10% Le[Y] expression, showed a weaker response, and PDX-201.1A-Cx organoids, with 5% Le[Y] expression, had no response to Le[Y] CAR T cells (Fig. 2d and Supplementary Fig. 1a, b). This is consistent with data from cell lines, with lysis of the Le[Y]-positive prostate cancer cells PC-3, DU-145, and 22Rv1, but not the Le[Y]-negative melanoma cell line MDA-MB435, when co-cultured with Le[Y] CAR T cells (Supplementary Fig. 1d, e). The mechanism of cell death in organoids involved the perforin/granzyme-dependent granule exocytosis pathway as granzyme B and/or perforin inhibitors markedly reduced cell death in PDX-287R and PDX-472M organoids (Fig. 2e).

Since Le[Y] can be targeted on primary prostate tumors in vitro, we next administered Le[Y] CAR T cells to mice bearing PDX-287R grafts derived from a patient with aggressive adenocarcinoma, as previously described[23]. PDX-287R has 70% Le[Y] expression by immunohistochemistry and 95% by flow cytometry (Fig. 1c and Supplementary Fig. 2a). As expected, CAR T cells administered alone did not inhibit tumor growth (Fig. 3a). As immune checkpoint blockade is often combined with CAR T cells in the setting of solid cancers[24], we combined CAR T therapy with nivolumab, an anti-PD-1 antibody that blocks its inhibitory interaction with its ligand, PD-L1. Despite the tumors expressing PD-L1 (Supplementary Fig. 2b, c), nivolumab had no effect on tumor growth alone or in combination with Le[Y] CAR T cells (Fig. 3b). We next trialed CAR T cells in combination with a single dose of the taxane-based chemotherapy docetaxel, or the platinum-based chemotherapy carboplatin, given a week prior to CAR T cell infusion. Docetaxel given alone, or with Le[Y] CAR T cells, had no effect on tumor growth (Fig. 3c). Carboplatin treatment alone or with ev-T cells induced an initial decrease in tumor burden that was sustained for up to 3 weeks before tumors grew back (Fig. 3d, e and Supplementary Fig. 2d). In contrast, carboplatin with CAR T cells caused tumors to regress to <1% of their starting volume, with no regrowth for 6 weeks post-treatment (Fig. 3d, e). Following Le[Y] CAR T cell plus carboplatin treatment only scattered tumor cells could be detected in 5 of the 6 residual grafts (Supplementary Fig. 2e), with phospho-histone H3 (pHH3) staining indicating greatly decreased proliferation compared to tumor cells in other treatment groups (Fig. 3f). Residual tumor cells lacked or had minimal Le[Y] expression (Supplementary Fig. 2f). However, treatment with carboplatin alone did not alter Le[Y] expression in tumors in vivo, as well as in organoids following short-term carboplatin treatment in vitro, suggesting that the lack of Le[Y] expression in residual tumors was not due to carboplatin treatment (Supplementary Fig. 2g–i). These data showed that pre-treatment with carboplatin could negate the poor therapeutic response of CAR T cells alone in PDX-287R.

To determine whether this combination therapy is effective in other PDX lines, we treated PDX-224R-Cx, a primary neuroendocrine cancer responsive to Le[Y] CAR T cells in vitro (Fig. 2d). Consistent with PDX-287R, Le[Y] CAR T cells alone did not alter tumor growth (Fig. 3g). Carboplatin treatment alone caused an initial regression in tumor burden that was sustained for a week before grafts grew back (Fig. 3g). Treatment with carboplatin plus Le[Y] CAR T cells significantly decreased tumor burden compared to no T cell control, carboplatin, Le[Y] CAR T cells alone and carboplatin with ev-T cells at the end of the treatment period (Fig. 3g). However, in contrast to PDX-287R, tumors continued to grow across the experimental period (Fig. 3g), suggesting that the response to carboplatin-CAR T cell therapy may depend on the effect of carboplatin in individual tumors.

To investigate this, we first analyzed T cell recruitment and activation, which is a major barrier to CAR T cell therapy in solid tumors. We assessed the localization of recruited Le[Y] CAR T cells in PDX-287R grafts by staining for the human T cell marker CD3. There was significant enrichment of Le[Y] CAR T cells at the tumor site 6 weeks after treatment with carboplatin, compared to all other treatment groups and controls (Fig. 4a, b), demonstrating that the carboplatin-induced long-term CAR T retention at the graft sites. To determine if CAR T cells

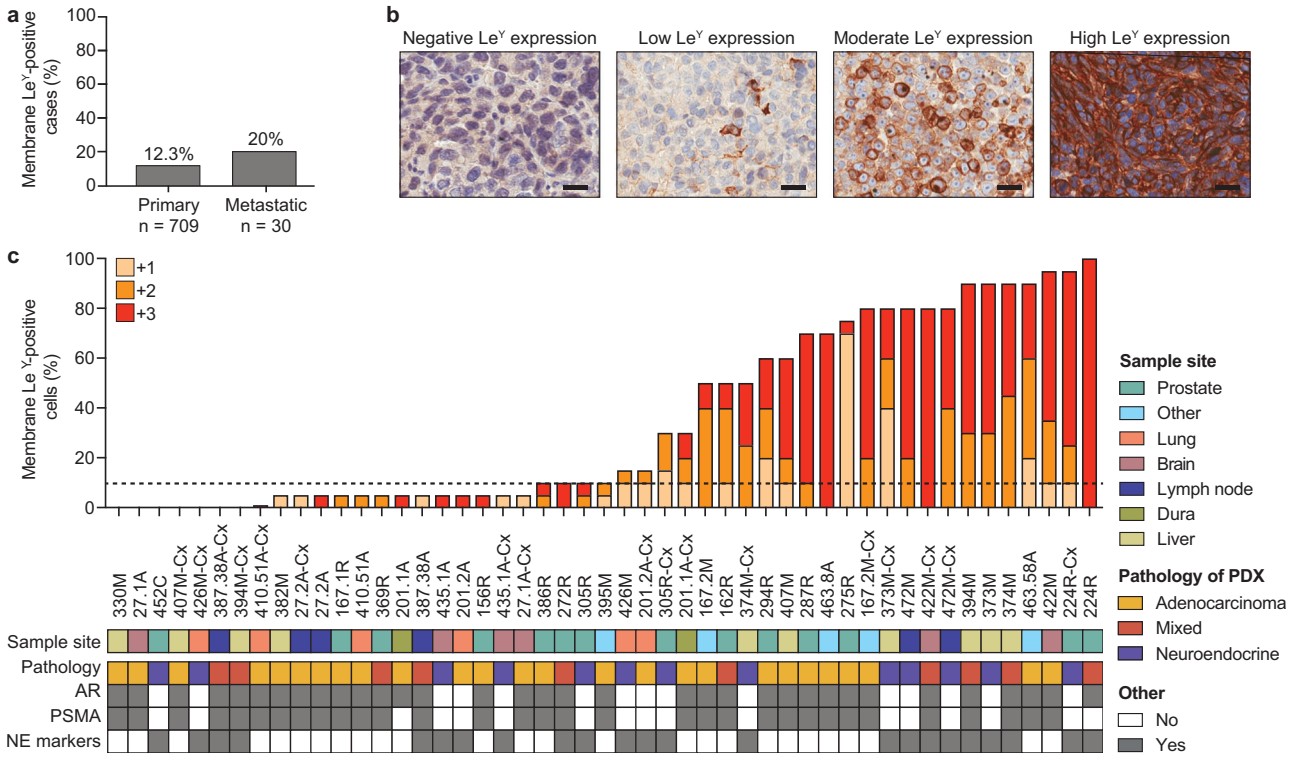

**Fig. 1 | Expression of Le^Y in human prostate cancer. a** The percentage of primary tumors (*n* = 709) or metastatic tumors (*n* = 30) positive for membrane Le^Y expression. Tumors were considered positive with ≥10% membrane Le^Y-positive cells. **b** Representative images of immunohistochemical staining of Le^Y in PDXs showing negative (0%), low (<10%), moderate (10–50%), and high (>50%) membrane Le^Y expression (scale bars = 25 μm). **c** The percentage of **c**ells positive for low intensity (+1; yellow), moderate intensity (+2; orange), and high intensity (+3; red) membrane Le^Y expression in PDXs from the MURAL cohort. Staining was repeated on three generations of PDX tissue, and results from the latest generation are shown. NE neuroendocrine. Source data are provided as a Source Data file.

homed directly to the tumor, or if infiltration was secondary to tumor shrinkage, we assessed Le^Y CAR T cell infiltration 48 h after infusion. Remarkably, we found high numbers of Le^Y CAR T cells in grafts 48 h after infusion following pre-treatment with carboplatin compared to grafts treated with CAR T cells alone (Fig. 4c, d), with concurrent accumulation of mouse F4/80⁺ macrophages (Fig. 4e, f). Using flow cytometry, we showed that the presence of CAR T cells in the tumor was accompanied by a decrease in CAR T cells in the spleen and peripheral blood, when compared to CAR T cells administered alone (Fig. 4g). Elevated expression of T cell activation markers CD25 and CD137, and the production of pro-inflammatory Th1 cytokines IL-2, TNF-α, and IFN-γ, at the tumor site were also consistent with heightened T cell activation when administered after carboplatin (Fig. 4h, i and Supplementary Fig. 3a–d).

In contrast to PDX-287R, similar analysis on PDX-224R-Cx grafts 48 h post-CAR T cell infusion demonstrated that carboplatin failed to increase the infiltration of CAR T cells into PDX-224R-Cx tumors (Fig. 4j). However, the limited CAR T cells that were present in the tumors were activated following carboplatin treatment (Fig. 4k, l and Supplementary Fig. 3e–h). Therefore, while TME-modulation with carboplatin increased the activation of Le^Y CAR T cells in PDX-224R-Cx, the response was restricted by a lack of CAR T cell infiltration into the tumor.

We next interrogated changes in the TME using flow cytometry seven days after carboplatin treatment, concurrent to the time of CAR T cell infusion. Carboplatin-treated PDX-287R grafts had a significant decrease in the proportion of EpCAM⁺ tumor epithelial cells 1 week following treatment compared to vehicle-treated grafts (Fig. 5a, b). This was accompanied with a significant increase in hFAS-positive tumor cells (Fig. 5c and Supplementary Fig. 4a), a death receptor associated with extrinsic apoptosis. In contrast, despite PDX-224R-Cx

initially responding to carboplatin treatment in vivo (Fig. 3g), by 1 week after treatment there was no change in the proportion of tumor cells or hFAS expression (Fig. 5d–f and Supplementary Fig. 4b).

To investigate the response of tumor cells to carboplatin treatment in more detail, RNA-sequencing was performed on FACs-isolated tumor cells from PDX-287R at 1 week post-treatment. Carboplatin resulted in the differential expression of 2017 genes in tumor cells, with 982 upregulated and 1036 downregulated genes (Fig. 5g and Supplementary Data 1). Consistent with the increase in FAS expression observed by flow cytometry, gene set enrichment analysis showed an enrichment for apoptotic pathways, with significant increased expression of the pro-apoptotic genes *FAS, BAX, BBC3, IFI6*, and *JUN* (Fig. 5h and Supplementary Fig. 4c). Gene set enrichment analysis also revealed enrichment of interferon response, JAK/STAT signaling, p53, and chemokine signaling pathways (Supplementary Fig. 4c). Previously, carboplatin has been shown to activate the cGAS/STING pathway, resulting in an increase in chemokine expression by lung tumor cells[22]. Here, we also found an increase in STING signaling, with an enrichment for the cytosolic DNA sensing pathway and increased expression of *STING1, STAT1*, and *STAT2* (Fig. 5i and Supplementary Fig. 4c). There was also a significant increase in the expression of genes involved in T cell chemotaxis, *CXCL10, CXCL11, CCL20* (Fig. 5j and Supplementary Data 1). This increase in chemokine expression may contribute to early recruitment of CAR T cells into the tumor. Collectively, carboplatin-induced cell death in tumor cells likely initiated a pro-inflammatory phenotype in prostate cancer PDXs.

At 1 week post-treatment, flow cytometry analysis also showed that carboplatin caused an influx of CD45⁺ immune cells into PDX-287R grafts (Fig. 5a and Supplementary Fig. 4d). The majority of immune cells in both control and treated grafts were F4/80⁺ cells of the myeloid monocytic lineage, and RNA seq of isolated immune cells

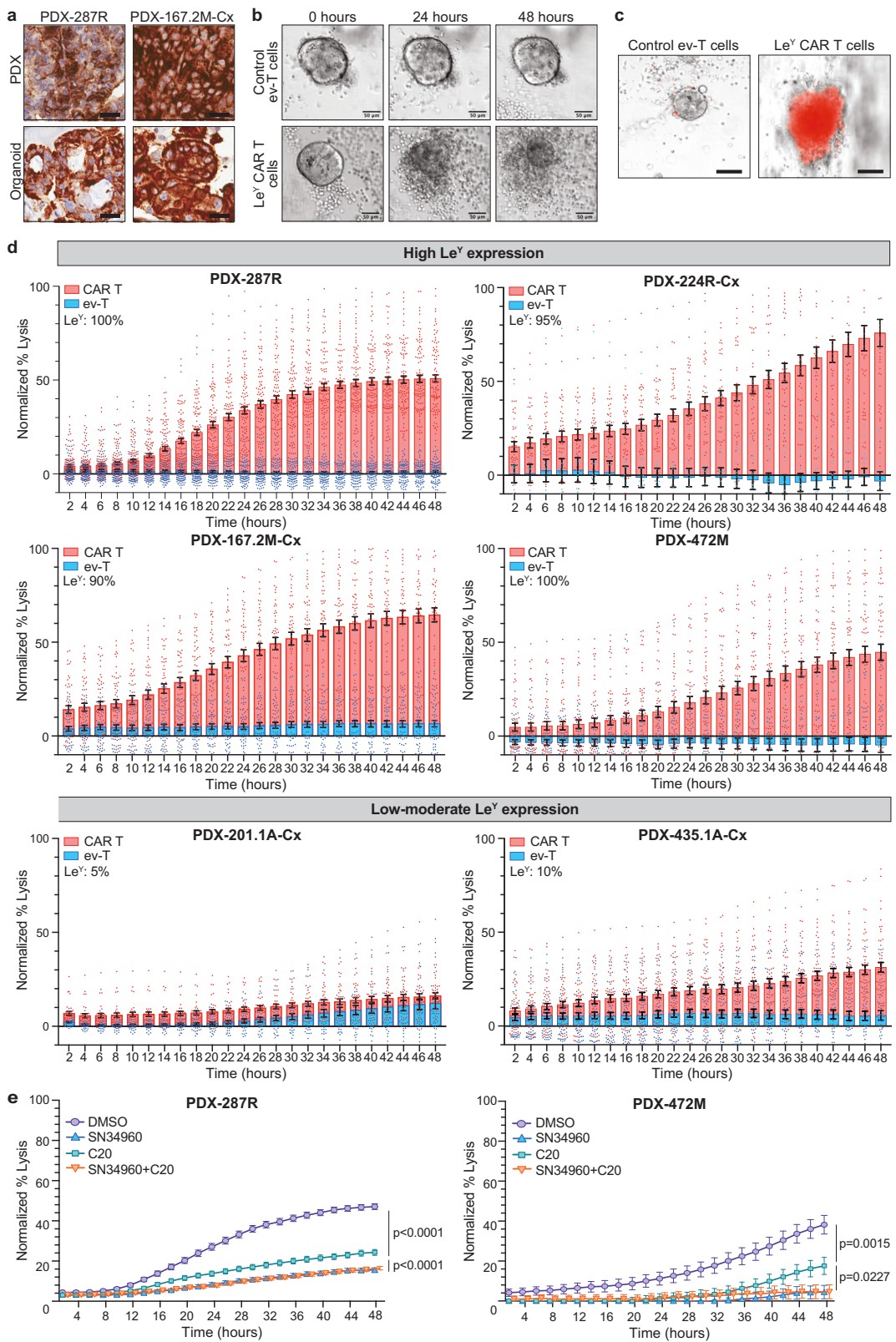

demonstrated no change in macrophage phenotype between control and treated grafts (Supplementary Fig. 4e–h and Supplementary Data 2). In contrast, in PDX-224R-Cx grafts there was no change in CD45⁺ immune cell infiltration following carboplatin treatment (Fig. 5d and Supplementary Fig. 4i). PDX-287R grafts treated with docetaxel also showed no change in tumor cell proportion and FAS expression at 1 week post-treatment, and a reduction in immune cell infiltration was

observed (Supplementary Fig. 4j–n). Therefore, a response to combination therapy with CAR T cells may depend on initial sensitivity to the chemotherapeutic agent, with tumor cell death at the time of CAR T cell infusion required to promote T cell recruitment into the tumor.

To further understand the long-term changes following carboplatin treatment in PDX-287R that may facilitate CAR T cell retention and the sustained response to Leʸ CAR T cells, we used scRNA-seq to

**Fig. 2 | On-target killing of prostate cancer organoids by Le^Y CAR T cells.**
**a** Representative immunohistochemical staining (*n* = 3 samples per PDX/organoid) of Le^Y in PDXs or organoids from PDXs (scale bars = 25 μm). **b** Representative images of Le^Y-positive organoids killed by infiltrating Le^Y CAR T cells compared to empty vector T (ev-T) cells (scale bars = 50 μm). **c** Propidium iodide (PI; red) uptake by organoids after 24-h culture with Le^Y CAR T cells, but not ev-T cells (scale bars = 50 μm). **b, c** Representative images of data presented in **d. d** PI staining of organoids with high or low-moderate Le^Y expression co-cultured with Le^Y CAR T cells (red) and ev-T cells (blue) for 48 h. Organoids were established from PDX-287R (*n* = 100 and 90 organoids co-cultured with CAR T and ev-T cells, respectively), PDX-224R-Cx (*n* = 42 and 15 organoids for CAR T and ev-T cells, respectively), PDX-167.2M-Cx (*n* = 50 organoids for CAR T or ev-T cells, respectively), PDX-472M (*n* = 53 and 46 organoids for CAR T and ev-T cells, respectively), PDX-435.1A-Cx (*n* = 50 organoids for CAR T or ev-T cells) and PDX-201.A-Cx (*n* = 33 and 31

organoids for CAR T and ev-T cells, respectively). Normalized percent lysis was calculated for CAR T or ev-T organoids as (MFI of organoid − min_mean)/(max_mean − min_mean) x 100, with MFI of Triton set as maximal lysis (max) and MFI of organoids alone as spontaneous lysis (min). The percentage membrane Le^Y-positive cells is shown for each organoid model. **e** Killing by Le^Y-positive CAR T cells is abrogated by granzyme and perforin inhibitors. Graphs show normalized percent lysis of organoids by CAR T cells over 48 h established from PDX-287R and treated with DMSO, perforin inhibitor (SN34960), granzyme B inhibitor (C20) and combined inhibition (*n* = 100 organoids per treatment), or established from PDX-472M and treated with DMSO (*n* = 54 organoids), SN34960 (*n* = 56 organoids), C20 (*n* = 60 organoids), and combined inhibition (*n* = 53 organoids). Data in **d** and **e** are shown as mean ± SEM. Significance was determined by two-tailed unpaired *t* test. Source data are provided as a Source Data file.

transcriptionally define immune and non-immune cell populations and anti-tumor innate immune signaling within the TME 3 weeks post-carboplatin treatment, when the response to carboplatin was maximal. Consistent with the previous experiment, a single dose of carboplatin initially induced a decrease in tumor burden before tumors grew back (Supplementary Fig. 5a). Indeed, by 3 weeks post-carboplatin treatment 97% (28/29) of grafts from across the two experiments had regressed from starting volume (Supplementary Fig. 5b). Given the tumor mass was considerably reduced by this timepoint, we expected a substantial reduction in tumor cells. This was demonstrated by scRNA-seq, where epithelial cells comprised only 2.1% of the cellular content of the graft, compared to 86.1% in vehicle control, and confirmed histologically (Fig. 6a, Supplementary Fig. 5c, and Supplementary Data 3). Unlike the 1 week time point, there was no significant difference in the expression of pro-apoptotic genes; however, *BAX* and *BAD* were still upregulated suggesting ongoing apoptosis (Supplementary Fig. 5d and Supplementary Data 4).

As the epithelial cell content was reduced in carboplatin-treated PDX tumors, there was a proportionate increase in mouse stroma (97.9%) compared to vehicle control (13.9%; Fig. 6a, Supplementary Fig. 5c, and Supplementary Data 3). This was also accompanied by changes in the relative populations of stromal cell types, with an increase in macrophages and monocytes, and fewer fibroblasts, endothelial and perivascular cells (Fig. 6b and Supplementary Data 3). UMAP (Uniform Manifold Approximation and Projection) density-based clustering analysis separated PDX-287R-associated mouse stromal cells, showing distinct clusters of stromal cell types with differential gene expression patterns between cells from carboplatin-treated and vehicle grafts (Fig. 6c). The gene expression profile of each cell type was cross-referenced to known cell population markers[25–30] to identify the cell types represented by each cluster in our analysis (Supplementary Fig. 5e–g). In contrast to the epithelial cells, carboplatin treatment led to increased expression of pro-survival factors (*Bcl2, Mcl1, Ier3, Ier5, Hmox1*) in stromal cell populations, thus preserving their viability and potentially providing a means of stabilizing and prolonging the anti-tumor response (Supplementary Fig. 5d, Supplementary Data 5–7).

In contrast to 1 week post-treatment, gene set enrichment analyses demonstrated that by 3 weeks following carboplatin treatment macrophages had undergone polarization to the pro-inflammatory M1, but not M2, phenotype, with upregulated of *Cd68, Cd86,* and *MHCII (H2-Aa, H2-Eb1,* and *H2-Ab1)* (Fig. 6d, e and Supplementary Data 5). Consistent with their M1 differentiation, the macrophages upregulated expression of the chemotactic factors *Ccl5* and *Cxcl10,* responsible for recruiting T lymphocytes (in this instance CAR T cells) into the tumor stroma (Fig. 6f).

The transcriptional profile of cancer-associated fibroblasts (CAFs) was also substantially altered 3 weeks after a single dose of carboplatin. The proportion of CAFs in carboplatin-treated grafts was markedly reduced (Fig. 6b, c), and there was a shift from a myofibroblast

(myCAF) to a pro-inflammatory phenotype (iCAF) (Fig. 6g and Supplementary Data 6). This involved the downregulation of genes encoding the immunosuppressive factors *Tgfß-1* and *Vegfa,* as well as the *Loxl* gene family, responsible for stabilizing the extracellular matrix (ECM; Fig. 6h and Supplementary Data 6). Following carboplatin treatment, CAFs increased their expression of numerous MMP genes (*Mmp2, Mmp3, Mmp13, Mmp14, Mmp27*); proteases involved in the degradation of the tumor-supporting matrix (Fig. 6h and Supplementary Data 6). Carboplatin-treated CAFs also upregulated the chemoattractant *Ccl2,* which has been associated with promoting monocyte and macrophage infiltration of the tumor site, and *Irf1* and *IL-1ß,* which enhance M1 polarization in macrophages (Fig. 6h and Supplementary Data 6). Thus, the carboplatin-induced CAF phenotype orchestrates a coordinated anti-tumorigenic response, which facilitates the polarization of macrophages to a pro-inflammatory M1 phenotype, enables infiltration of phagocytes to the tumor site, and alters the ECM to promote immune cell trafficking. Taken together, carboplatin induces multiple changes that could overcome the immunosuppressive TME.

CAR T cells gain entry to the tumor site through the process of transendothelial migration, and tumor-associated high endothelial venule (TA-HEV) cells have been described to facilitate this trafficking[30]. Three weeks post-carboplatin treatment there was a reduction in the proportion of endothelial cells in the PDXs (Fig. 6b). However, multiple phenotypic changes were observed. Firstly, endothelial cells showed decreased expression of mouse *Angpt2* and *Egln1* genes, which typically promote tumor cell metastasis and the dysregulation of vascular stability (Fig. 6i and Supplementary Data 7). Furthermore, carboplatin-treated endothelial cells had increased expression of *Sele* and *Selp,* which encode key selectins involved in the initial "capture and rolling" of leukocytes along the endothelial layer (Fig. 6i and Supplementary Data 7). Downstream in the signaling pathway, carboplatin-treated endothelial cells also upregulated endothelial adhesion molecules *Icam1,* as well as *Vcam1* gene expression, both of which may enhance that typify the TA-HEV phenotype and could be expected to enhance CAR T cells traversal through the endothelial layer (Fig. 6i and Supplementary Data 7).

Collectively, we describe a cascade of anti-tumorigenic alterations in cell content and gene expression in response to low-dose carboplatin in PDX-287R that demonstrate how carboplatin effectively primes the TME and promotes Le^Y CAR T efficacy (Fig. 7). In the absence of carboplatin sensitivity, including upregulation of hFAS and immune cell infiltration, CAR T cell entry to the tumor is limited and the response to Le^Y CAR T cell treatment is dampened.

## Discussion
Collectively, these studies demonstrate that Le^Y CAR T cells are effective at targeting prostate cancer cells when co-cultured in vitro and can eliminate a prostatic PDX tumor in vivo when preceded by a single dose of carboplatin. Mechanistically, carboplatin induced a

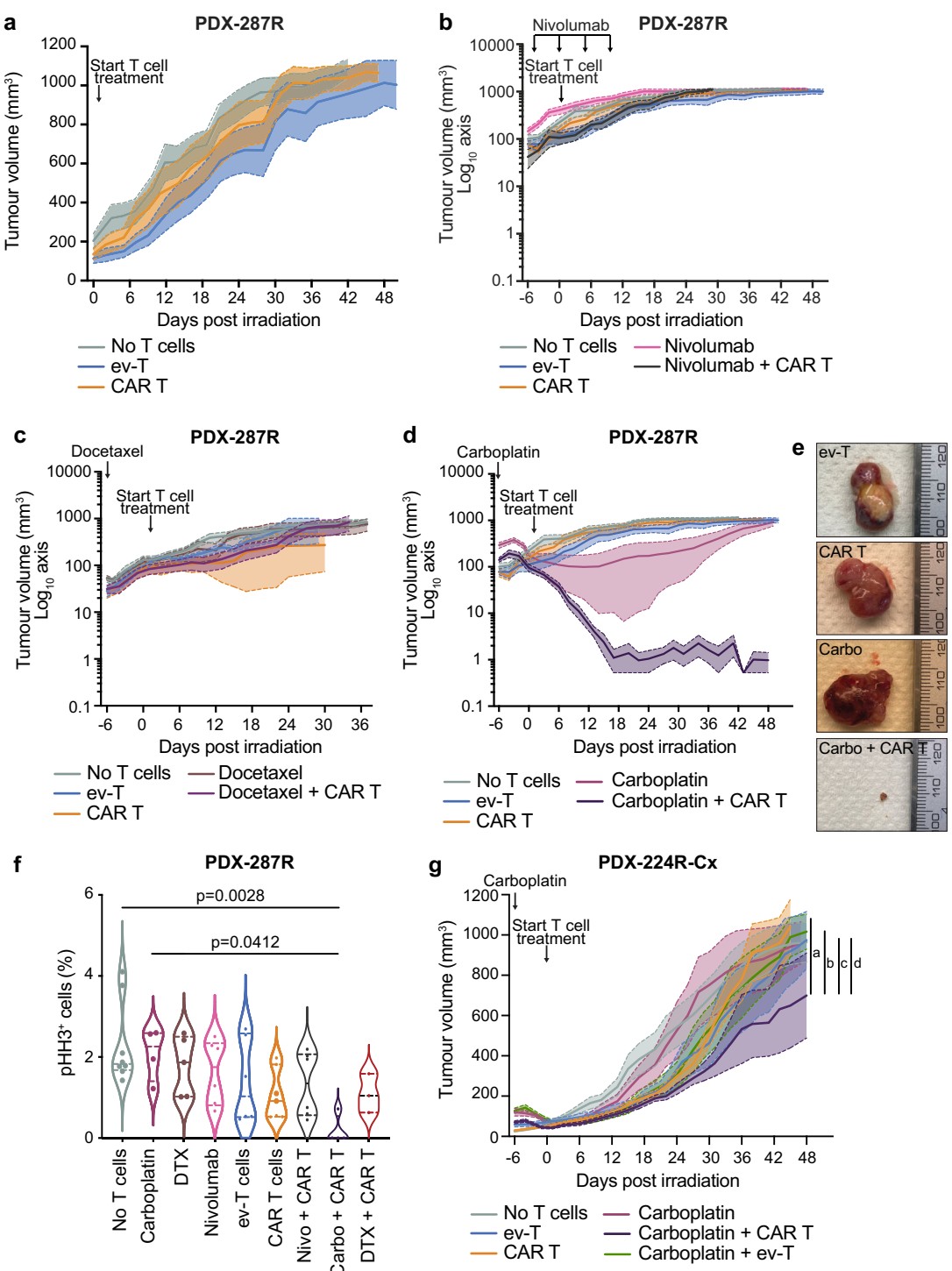

cascade of events following the initial apoptosis of tumor cells, resulting in a pro-inflammatory, anti-tumor immune response that enables CAR T cells to enter the tumor site and exert cytotoxic effector functions on Le$^Y$-expressing tumor cells. Failure of carboplatin to induce apoptosis at the time of CAR T cell infusion and decreased T cell recruitment into the tumor dampens, but does not abolish, the response to treatment. Therefore, carboplatin improves the efficacy of CAR T cell treatment, with the extent of the response dependent on changes induced within the tumor.

Trafficking of CAR T cells to the tumor is a major barrier to CAR T cell therapy in all cold, immune excluded solid tumors, and our data further supports this notion. In PDX-287R, carboplatin treatment

initially induced apoptosis within the tumor cells, accompanied by a pro-inflammatory response, increased cGAS-STING signaling and increased expression of chemokines, including CXCR3 ligands for T cell chemotaxis. At this time, there was an influx of mouse myeloid cells into the tumor, and enhanced recruitment and activation of human CAR T cells following carboplatin treatment. In contrast, carboplatin failed to increase tumor cell apoptosis, myeloid cell recruitment and T cell infiltration in PDX-224R-Cx. Interestingly, the T cells present within the PDX-224R-Cx tumors still showed increased activation following carboplatin, presumably accounting for the decrease in tumor burden in carboplatin-Le$^Y$ CAR T cell treated grafts compared to treatment with Le$^Y$ CAR T cells alone. Collectively, this demonstrates

**Fig. 3 | Low-dose chemotherapy enhances anti-tumor efficacy of Le$^Y$ CAR T cells in vivo. a** Tumor volume (mean ± SEM) of PDX-287R grafts following treatment with no T cell control ($n = 21$ grafts), Le$^Y$ CAR T cells ($n = 17$ grafts) or control empty vector T (ev-T; $n = 17$ grafts) cells alone. Data from three independent experiments. **b** Tumor volume (mean ± SEM) of PDX-287R grafts following treatment with no T cell control ($n = 15$ grafts), Le$^Y$ CAR T cells ($n = 12$ grafts), ev-T cells ($n = 12$ grafts), nivolumab (200 μg/dose, 4 doses, $n = 13$ grafts), and nivolumab with Le$^Y$ CAR T cells ($n = 8$ grafts). Data from two independent experiments. **c** Tumor volume (mean ± SEM) of PDX-287R grafts following treatment with no T cell control ($n = 6$ grafts), Le$^Y$ CAR T cells ($n = 5$ grafts), ev-T cells ($n = 5$ grafts), docetaxel (10 mg/kg, 1 dose; $n = 6$ grafts) and docetaxel with Le$^Y$ CAR T cells ($n = 6$ grafts). Data from 1 experiment. **d** Tumor volume (mean ± SEM) of PDX-287R grafts following treatment with no T cell control ($n = 15$ grafts), Le$^Y$ CAR T cells ($n = 12$ grafts), ev-T cells ($n = 12$ grafts), carboplatin (50 mg/kg, 1 dose; $n = 17$ grafts), and carboplatin with Le$^Y$ CAR T cells ($n = 10$ grafts). Data from two independent experiments. **e** Representative PDX-287R grafts after treatment with ev-T cells ($n = 12$ grafts), CAR T cells ($n = 12$ grafts), carboplatin (carbo; $n = 17$ grafts), and carboplatin with CAR T cells (carbo + CAR T cells; $n = 10$ grafts). **f** Quantification of pHH3 immunohistochemistry in PDX-287R grafts 5–6 weeks post-treatment with no T cell control ($n = 8$ grafts), carboplatin ($n = 4$ grafts), docetaxel (DTX; $n = 5$ grafts), nivolumab ($n = 6$ grafts), ev-T cells ($n = 6$ grafts), CAR T cells ($n = 7$ grafts), nivolumab (nivo) + CAR T cells ($n = 6$ grafts), carbo + CAR T cells ($n = 4$ grafts), and DTX + CAR T cells ($n = 3$ grafts). Statistical significance determined by one-way ANOVA with post-hoc Tukey's test. **g** Tumor volume (mean ± SEM) of PDX-224R-Cx grafts following treatment with no T cell control ($n = 12$ grafts), Le$^Y$ CAR T cells ($n = 6$ grafts), ev-T cells ($n = 8$ grafts), carboplatin (50 mg/kg, 1 dose; $n = 14$ grafts), carboplatin with Le$^Y$ CAR T cells ($n = 6$ grafts), and carboplatin with ev-T cells ($n = 9$ grafts). Data from two independent experiments. Compared to PDX-287R, PDX-224R-Cx had decreased sensitivity to carboplatin treatment alone, and a reduced response to carboplatin-CAR T cell combination treatment. Significance was determined by linear mixed model analysis at end of treatment with a test of simple main effects to compare between treatment groups: **a** CAR T cells vs carboplatin + CAR T cells, $p = 0.015$; **b** carboplatin + ev-T cells vs carboplatin + CAR T cells, $p = 0.044$; **c** no T cells vs carboplatin + CAR T cells, $p = 0.008$; **d** carboplatin vs carboplatin + CAR T cells, $p = 0.028$. Source data are provided as a Source Data file.

---

that pro-inflammatory changes induced by carboplatin are mandatory to enable the CAR T cells to gain unfettered access to the Le$^Y$-positive tumor cells and eliminate the tumor.

Whilst tumor cell sensitivity to carboplatin is clearly a feature, the transcriptomic changes within the TME following carboplatin treatment strongly suggest that pro-inflammatory effects are not solely dependent on cancer cell death. A single dose of carboplatin induced a cascade of events to sustain a pro-inflammatory microenvironment and T cell persistence. The increase in macrophages initially observed at 1 week post-treatment was sustained, with a shift to the M1 pro-inflammatory phenotype by 3 weeks post-treatment. Indeed, carboplatin is known to activate mouse macrophages[31,32], and consequently, the increased production of macrophage-derived factors, *Ccl5* and *Cxcl10*, likely mediated the recruitment of T cells through known mechanisms, including TCR3 receptor recruitment[33]. The evidence from our animal studies strongly suggests that the myeloid compartment plays a key role in the mechanism of action of carboplatin. Whether the response will be similar in humans is unknown, but it is of high relevance to the clinical translation of this combination therapy.

In addition to the activation of macrophages, carboplatin induced a range of downstream modifications to the TME, including the induction of an iCAF phenotype, involving the downregulation of immunosuppressive genes such as *Tgfβ-1* and *Vegfa*, and decreased expression of *LOXL* genes and increased expression of *MMP* genes, leading to ECM degradation. We also demonstrated the presence of tumor-associated high endothelial venule (TA-HEV) cells that are known to effectively traffic lymphocytes to tumors[30]. Collectively, this cascade of dynamic alterations to the TME (Fig. 7), observed after carboplatin treatment, resulted in the induction of a pro-inflammatory microenvironment and provides a mechanism by which human T cell efficacy is mediated in this immune-compromised setting.

The chemotherapeutic agents docetaxel and cyclophosphamide have been shown to induce immunogenic cell death, dampen the suppressive TME, and improve the efficacy of immunotherapy, including CAR T cells and checkpoint blockade, in preclinical studies for prostate cancer[5,7,21]. Furthermore, docetaxel is currently in clinical trials in combination with immunotherapy for chemotherapy-naïve patients with advanced prostate cancer, where it is well-tolerated and showing anti-tumor activity[34]. Here, we demonstrate that the response to chemotherapies as a modulating agent is likely dependent on the response of individual tumors, with TME modifications at the transcriptional level likely playing a major role in CAR T cell infiltration and tumor response. In contrast to carboplatin, docetaxel failed to induce apoptosis and myeloid cell infiltration in PDX-287R at 1 week post-treatment, and had no effect on tumor growth when combined with CAR T cells. It is likely that a multimodal approach is required to enhance the efficacy of immunotherapy in prostate cancer, and a pro-inflammatory gene signature induced by agents such as carboplatin may be helpful in identifying patients that may benefit from CAR T cell therapy.

Interestingly, PDX-287R has a mutation in the mismatch repair gene mutS homologue 2 (*MSH2*), which may be associated with response to immune checkpoint inhibitors in a range of solid tumors[35]. This may have contributed to the response seen here and needs to be investigated in other prostate tumors with mismatch repair deficiency. The sequencing and timing of combination therapies may also influence response, and future studies are required to optimize their roles in enhancing CAR T cell therapy, for example, the use of real-time analyses to provide more dynamic insights into the spectrum of responses across time.

In this study, the action of carboplatin was assessed in immunosuppressed NSG mice, which lack T cells, B cells, and natural killer cells. This model allows us to compare the response of individual patient tumors to treatment, and thus it is more reflective of tumor heterogeneity seen in the clinic. In addition, a potentially unappreciated benefit of this immunosuppressed model is that it allows the focus to be on the interaction between the tumor and human CAR T cells without a cognate effector immune response being induced through the murine lymphoid compartment. This lack of lymphoid cells was advantageous as it did not obscure the major shift we saw in the phenotypes of host myeloid cells, fibroblasts, ECM, and endothelial cells in response to carboplatin. This is likely to be a critical facet of the mechanism of action and is highly relevant to the human setting. However, this model is limited by the lack of crosstalk between immune cells within the TME, including the dynamic interaction between immune activation and suppression, and future studies need to be extended into an immunocompetent syngeneic setting.

In this study, we examined the effect of bulk transduced T cells using IL-2; however, future studies should also consider the effectiveness of selected CAR T cell products with defined subsets of T cells that possess increased memory capability[36]. Evidence is emerging to show that maintaining CAR T cells in an early, less-differentiated state in vitro results in superior persistence, proliferation, and anti-tumor effects in vivo[36,37]. Although effector memory T cells have strong cytolytic capacity, less differentiated T cells, such as stem cell memory T cells, are critical for in vivo expansion and long-term persistence[37,38].

Le$^Y$ is not a predicted target antigen for prostate cancer treatment. Unlike other cell-surface prostate adenocarcinoma targets, such as prostate stem cell antigen (PSCA), prostate-specific membrane antigen (PSMA), and six-transmembrane epithelial antigen of the prostate (STEAP-1)[7,39], Le$^Y$ is expressed on a broad spectrum of tumors. In general, expression of PSCA, PSMA and STEAP-1 is limited to AR-

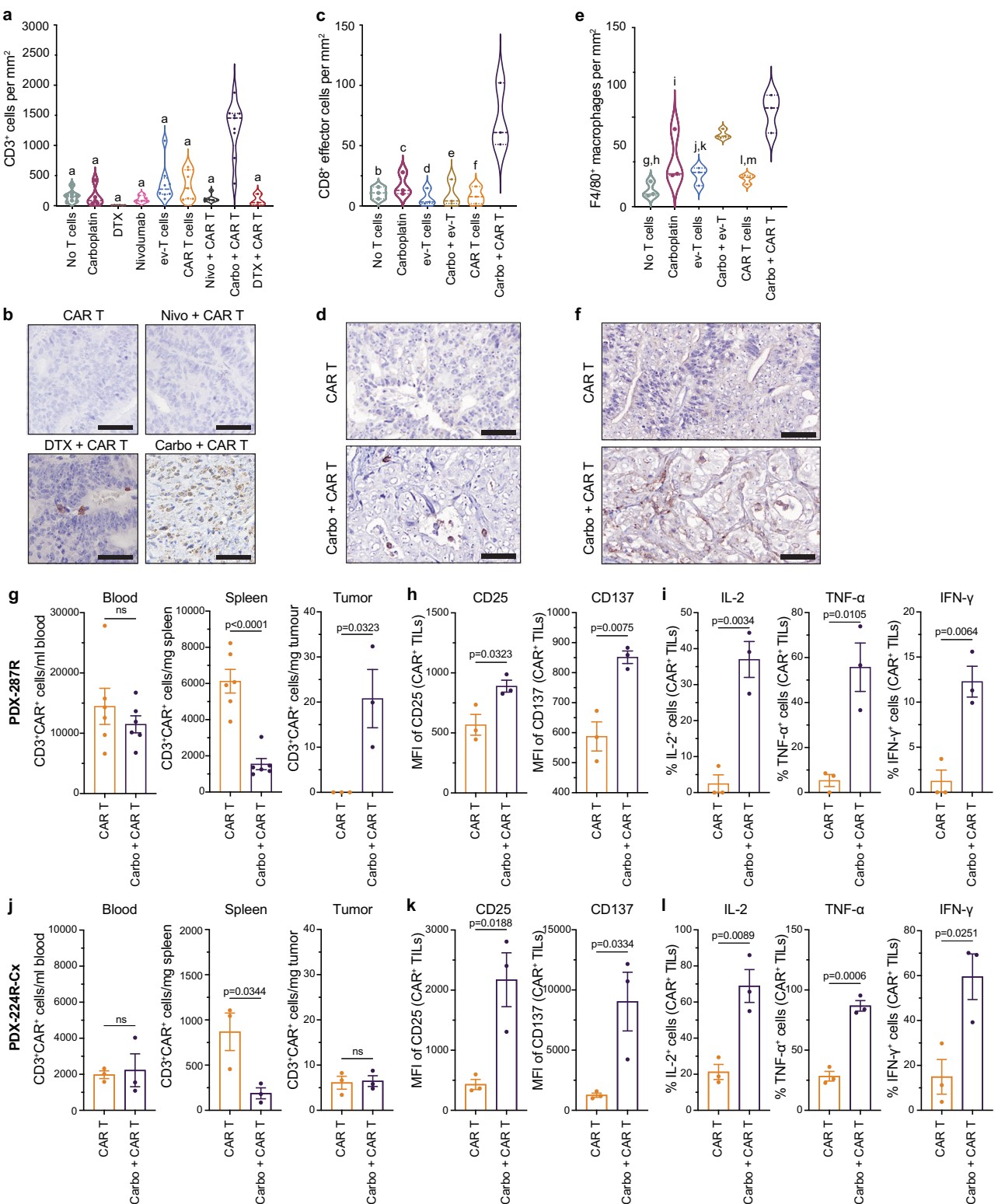

positive adenocarcinoma tumors and is rarely detected in AR-null or neuroendocrine prostate cancer, limiting their clinical application. Here, we demonstrate that $Le^Y$ expression is detected in both adenocarcinoma and neuroendocrine prostate cancer, and is independent of AR or PSMA, and thus $Le^Y$ CAR T cells could be applicable for the treatment of a diverse range of prostate tumors. Here, PDX-224R, a de novo neuroendocrine tumor, had a weaker response to combination therapy compared to PDX-287R. However, this is more likely dependent on chemotherapy response given that both adenocarcinoma and neuroendocrine tumors were sensitive to $Le^Y$ CAR T cells in vitro. A larger panel of PDXs is required to determine whether this response is phenotype dependent. This is clinically relevant as $Le^Y$ CAR T cells may provide another treatment option for patients who would not benefit from AR- or PSMA-directed therapies, or who had progressed following these therapies.

Overall, carboplatin induces a cascade of changes within the tumor to increase the efficacy of CAR T cell treatment, with the extent of the response dependent on TME modifications to promote T cell

**Fig. 4 | Carboplatin increases CAR T cell accumulation and activation in PDX-287R. a, b** Number of CD3+ T cells per mm², based on immunohistochemistry (**a**), and representative immunohistochemistry images of CD3+ T cells (**b**) in PDX-287R grafts 5–6 weeks post-treatment with no T cell control ($n = 8$ grafts), carboplatin ($n = 5$ grafts), docetaxel (DTX; $n = 6$ grafts), nivolumab ($n = 7$ grafts), ev-T cells ($n = 8$ grafts), CAR T cells ($n = 7$ grafts), nivolumab (nivo) + CAR T cells ($n = 6$ grafts), carboplatin (carbo) + CAR T cells ($n = 9$ grafts), and DTX + CAR T cells ($n = 3$ grafts). **c, d** Number of CD8+ effector T cells per mm², based on immunohistochemistry (**c**), and representative immunohistochemistry images of CD8+ effector T cells (**d**) in PDX-287R grafts after 48 h post-adoptive transfer with CAR T cells ($n = 3$ grafts/ treatment). **e, f** Number of F4/80+ mouse macrophages per mm², based on immunohistochemistry (**e**), and representative immunohistochemistry images of F4/80+ mouse macrophages (**f**) in PDX-287R grafts after 48 h post-adoptive transfer of CAR T cells ($n = 3$ grafts/treatment). **g–l** Quantification and activation of CAR T cells in mice bearing PDX-287R grafts (**g–i**) and PDX-224R-Cx grafts (**j–l**) 48 h post-adoptive transfer of CAR T cells by flow cytometry. Compared to PDX-287R, PDX-224R-Cx had decreased sensitivity to carboplatin treatment alone, and a reduced response to carboplatin-CAR T cell combination treatment. **g, j** Number of

CD3+CAR+ T cells extracted from blood, spleen, and tumor (TILs) in mice bearing PDX-287R (**g**) and PDX-224R-Cx (**j**; $n = 6$ mice/treatment for blood and spleen, $n = 3$ grafts/treatment for tumors). **h, k** Geometric mean fluorescence intensity (MFI) of CD25 and CD137 on TILs activated by anti-idiotype antibody of Le$^Y$ CAR in PDX-287R (**h**) and PDX-224R-Cx (**k**; $n = 3$ tumors/treatment). **i, l** Percentage of IL-2-, TNF-α-, and IFN-γ-positive TILs activated by anti-idiotype antibody of Le$^Y$ CAR using intracellular staining in PDX-287R (**i**) and PDX-224R-Cx (**l**; $n = 3$ tumors/treatment). Statistical significance in violin plots (**a, c, e**) was determined by one-way ANOVA with post-hoc Tukey's test: **a** all groups vs carbo + CAR T, $p < 0.0001$; **b** no T cells vs carbo + CAR T, $p = 0.0014$; **c** carboplatin vs carbo + CAR T, $p = 0.0033$; **d** ev-T cells vs carbo + CAR T, $p = 0.0008$; **e** carbo + ev-T vs carbo + CAR T, $p = 0.0011$; **f** CAR T cells vs carbo + CAR T, $p = 0.0010$; **g** no T cells vs carbo + ev-T, $p = 0.0048$; **h** no T cells vs carbo + CAR T, $p = 0.0003$; **i** carboplatin vs carbo + CAR T, $p = 0.0165$; **j** ev-T cells vs carbo + ev-T, $p = 0.0403$; **k** ev-T cells vs carbo + CAR T, $p = 0.0018$; **l** CAR T cells vs carbo + ev-T, $p = 0.0240$; **m** CAR T cells vs carbo + CAR T, $p = 0.0011$. Data in **g–l** represents the mean ± SEM, and the significance was determined by two-tailed unpaired $t$ test. ns not significant. Scale bars = 50 μm (**b, d, f**). Source data are provided as a Source Data file.

infiltration, activation and persistence. These data provide preclinical proof-of-concept evidence for the use of carboplatin as a modulating agent for CAR T cell immunotherapy in the treatment of prostate cancer.

## Methods

### Patient-derived xenografts

Serially-transplantable PDXs were established and characterized by MURAL, as previously described[23]. Original specimens of localized prostate cancer tissue were collected from patients undergoing radical prostatectomy or transurethral resection of the prostate, and metastatic prostate cancer tissue was obtained from biopsy, palliative surgery, or rapid autopsy through the CASCADE program[40]. Informed, written consent was obtained from patients prior to tissue collection according to human ethics approval from the Cabrini Institute (03-14-04-08), Monash University (1636), and Peter MacCallum Cancer Centre (15/98, 97_27), and patient details, including age and treatment history, have been previously published[23]. All PDX experiments were performed in 6–8-week-old male non-obese diabetic severe-combined immune-deficient (NSG) mice (RRID:IMSR JAX:005557)[41]. NSG mice were purpose bred at Monash Animal Research Laboratories (Monash University, Monash Breeding Colony approval number MMCA 209/ 25BC and 15160). All animal care and procedures for the establishment and maintenance of PDXs were performed in accordance with Monash University animal ethics approvals (MARP/2014/085, MARP/2014/119, MARP/2016/016, 17963, 28911). All mice were bred and housed in a PC2-specific pathogen-free facility under controlled temperature (22 °C) and lighting (12:12 h light-dark cycle), and were fed a chow diet *ad libitum*. Experimental mice and control mice were co-housed. NSG mice either had a 5 mm testosterone pellet implanted subcutaneously to supplement host testosterone levels, or were castrated to mimic androgen deprivation. Routine validation of PDX authenticity included DNA and RNA-sequencing, STR profiling and pathological assessment including biomarker expression for androgen receptor (AR), prostate specimen membrane antigen (PSMA) and the neuroendocrine markers CD56, synaptophysin and chromogranin A, as previously described[23]. Genomic alterations in PDXs, determined by targeted DNA sequencing, was previously published[23].

### Immunohistochemistry for Lewis Y

Le$^Y$ expression was analyzed in 709 primary prostate cancer specimens on the Case PSA Progression tissue microarray (obtained from the Prostate Cancer Biorepository Network) and 91 metastatic prostate cancer specimens from 32 patients in our own clinical collection. Le$^Y$ expression was also analyzed in 49 PDXs obtained from MURAL derived from 14 high-risk localized and 35 metastatic tumors.

Detection of the Le$^Y$ antigen was performed by immunohistochemistry on formalin-fixed tissue using a humanized monoclonal IgG1 antibody m3S193 (1.9 μg/ml; provided by Ludwig Institute for Cancer Research) on the Dako Autostainer Link 48 with EnVison FLEX, High pH Link visualization system reagents[42]. The staining protocol was adapted and validated from a previously published study[43]. Briefly, antigen retrieval was performed at 10 °C for 20 min, peroxidase blocking for 10 min, antibody or negative control reagent for 30 min at 1.9 μg/ml, EnVision Flex HRP polymer for 30 min, DAB for 5 min and finally sections were counterstained using Mayer's Hematoxylin. Normal human lung, NSCLC squamous, and adenocarcinoma control tissues were used for staining quality control throughout the study.

### Establishment of organoids from PDX tissues

Organoids were established from PDX tissues as previously described[23,44]. Fresh or frozen-thawed single PDX cells were seeded in growth factor reduced, phenol red-free, ldEV-free Matrigel (Corning), with $1$–$1.5 \times 10^5$ cells seeded per 20 μl of Matrigel in 48 well plates. Organoids were cultured in organoid media, consisting of advanced DMEM/F-12 media (Thermo Fisher) containing 1% penicillin-streptomycin, 2 mM Glutamax, 1 nM DHT, 1.25 mM N-acetylcysteine, 50 ng/ml EGF, 500 nM A83-01, 10 mM nicotinamide, 10 μM SB202190 (Sigma), 2% B27 (Life Technologies), 100 ng/ml noggin (Peprotech), 10 ng/ml FGF10 (VWR), 5 ng/ml FGF2, 1 μM prostaglandin E2 (Tocris), and 10% R-spondin 1 conditioned media. 10 μM Y-27632 dihydrochloride (Selleck Chemicals) was added to culture medium during organoid establishment and the first subsequent passage. To assess Le$^Y$ expression, organoids were fixed in 10% formalin for at least 1 h, and then embedded in paraffin before performing immunohistochemistry for Le$^Y$, as described above.

### CAR T cell production

The anti-Le$^Y$ scFv-CD3ζ-CD28 CAR construct and retroviral packaging cell line PG13-Le$^Y$ CAR was used in this study as described previously[10]. Human PBMCs were isolated from normal donor buffy coats (Australian Red Cross Blood Service) by density gradient centrifugation (Ficoll-Paque, GE Healthcare Life Science). PBMCs were stimulated with anti-human CD3 (OKT3 30 ng/ml, Miltenyi) and 600 units/ml IL-2 (Peprotech) for 48 h and were transduced on day 3 and 4 with viral supernatant produced by PG13-Le$^Y$ CAR cells on a Retronectin (Takara Bio, Otsu, Japan) coated 6-well plates as per the manufacturer's instructions. After 4 h incubation at 37 °C, viral supernatant was removed, and $2.5 \times 10^6$ T cells in 5 ml of fresh retroviral supernatant with IL-2 (600 units/ ml) were added. 24 h after the second transduction, T cells were transferred to fresh RPMI 1640 (Gibco) supplemented with 10% FBS, penicillin/streptomycin,

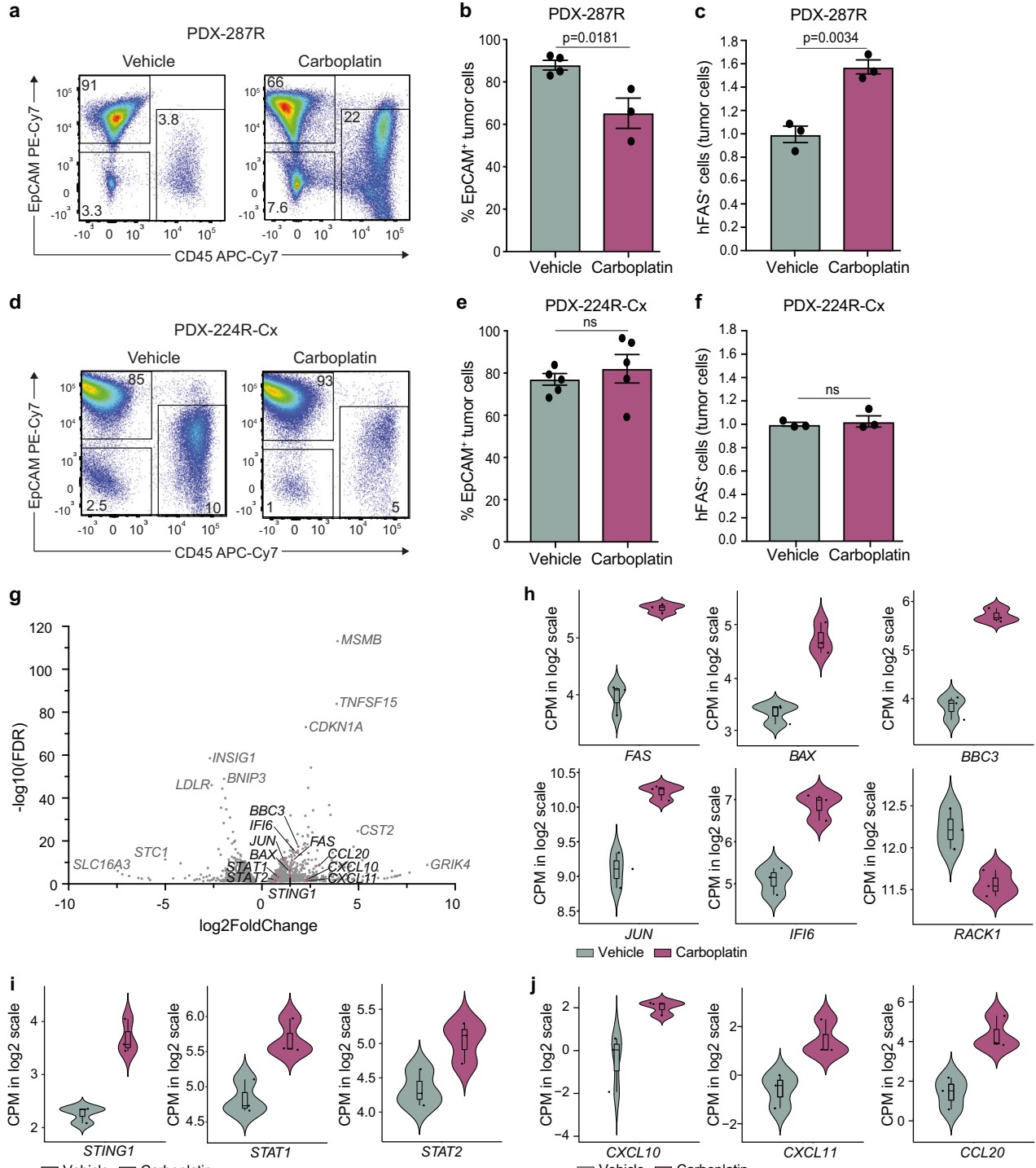

**Fig. 5 | Carboplatin induces tumor cell death and immune cell infiltration in PDX-287R, but not PDX-224R-Cx. a–f** Flow cytometry analyses of PDX-287R and PDX-224R-Cx 1 week following treatment with carboplatin (50 mg/kg, 1 dose). Compared to PDX-287R, PDX-224R-Cx had decreased sensitivity to carboplatin treatment alone, and a reduced response to carboplatin-CAR T cell combination treatment. **a** Representative flow cytometry plots showing proportions of EpCAM+ epithelial tumor cells, CD45+ immune cells and EpCAM-CD45- stromal cells in PDX-287R following vehicle (*n* = 4 grafts) and carboplatin treatment (*n* = 3 grafts). **b** The proportion of EpCAM+ epithelial tumor cells in PDX-287R grafts following vehicle (*n* = 4 grafts) and carboplatin treatment (*n* = 3 grafts). **c** The proportion of hFAS+ tumor cells, normalized to vehicle control, in PDX-287R grafts (*n* = 3 grafts/treatment). **d** Representative flow cytometry plots showing proportions of EpCAM+ epithelial tumor cells, CD45+ immune cells and EpCAM-CD45- stromal cells in PDX-

224R-Cx grafts following vehicle (*n* = 5 grafts) and carboplatin treatment (*n* = 5 grafts). **e** The proportion of EpCAM+ epithelial tumor cells in PDX-224R-Cx grafts (*n* = 5 grafts). **f** The proportion of hFAS+ tumor cells, normalized to vehicle control, in PDX-224R-Cx grafts (*n* = 3 grafts/treatment). **g** Volcano plot of genes significantly upregulated or downregulated, based on RNA seq, in tumor cells from PDX-287R 1 week following carboplatin treatment. **h–j** Violin plots showing expression of genes involved in apoptosis (**h**), cGAS-STRING signaling (**i**), and T cell chemotaxis (**j**) in PDX-287R tumor cells 1 week after treatment with vehicle or carboplatin (*n* = 3 samples/treatment). Data in **b**, **c**, **e**, **f** represents mean ± SEM, and the significance was determined by two-tailed unpaired *t* test. ns not significant. Box plots in **h–j** show the first to third quartile with median, whiskers show the minimum and maximum. Source data are provided as a Source Data file.

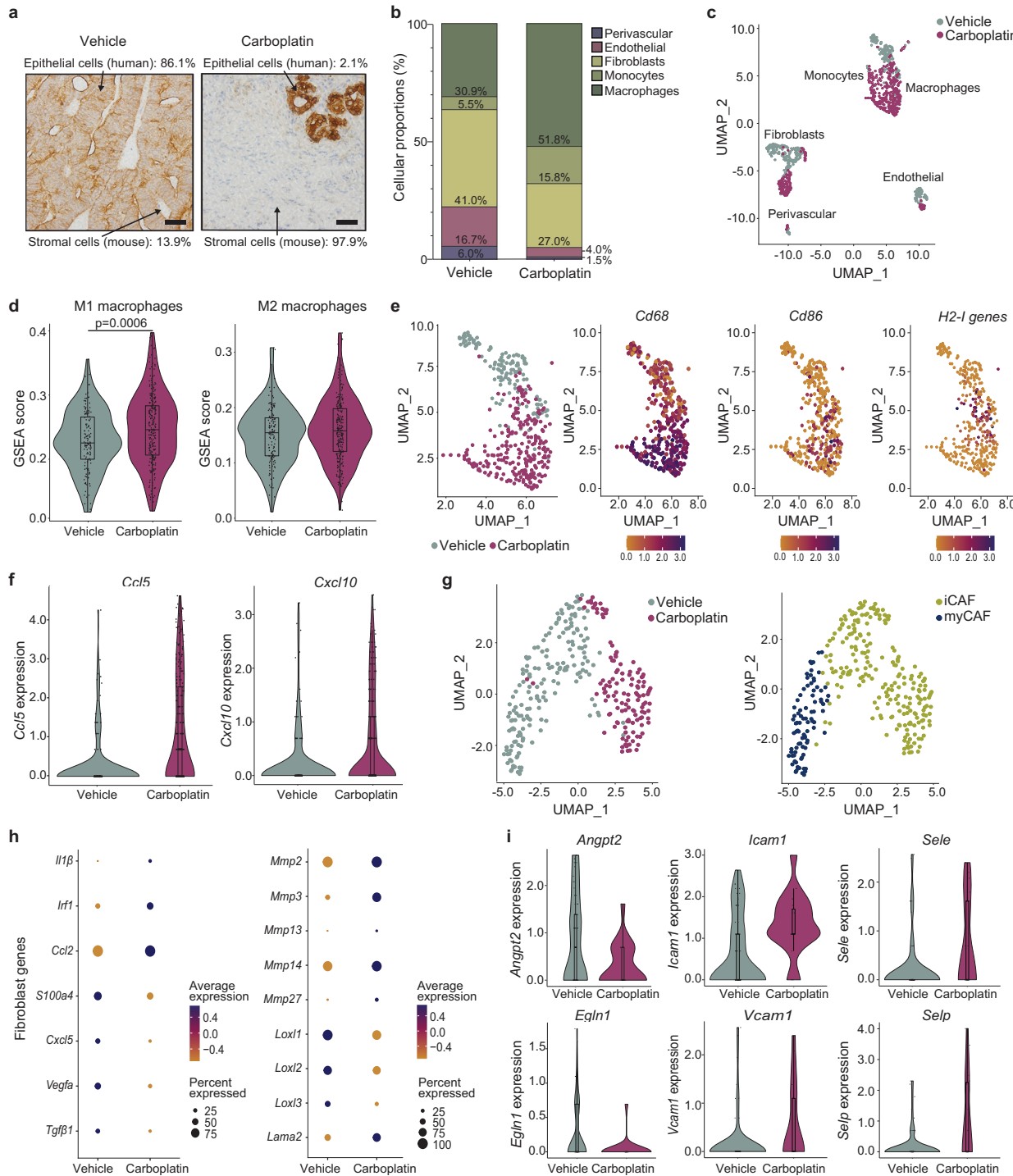

**Fig. 6 | Carboplatin induces a pro-inflammatory immune response, effectively priming the TME to facilitate Le$^\gamma$-specific CAR T cell infiltration in PDX-287R.**
**a** Representative images of residual PDX tumors 3 weeks post a single dose of carboplatin treatment (50 mg/kg; $n = 12$ grafts) or vehicle ($n = 7$ grafts) using immunohistochemistry for the human epithelial cell marker CK8/18 (scale bars = 50 μm). Cellular composition (%) of human epithelium and murine stroma in each treatment group, as determined by scRNAseq, is also shown. **b** Proportions of infiltrative cell types within carboplatin-treated ($n = 5$ samples) and vehicle control ($n = 1$ sample) stromal tissue (%). **c** UMAP-defined stromal cell populations. **d** Violin plots showing GSEA score enriched proportions of M1 and M2 macrophages within carboplatin-treated ($n = 245$ cells) and vehicle control tissue ($n = 135$ cells). GSEA statistical and empirical evaluation were used to yield a normalized enrichment score (NES; significance level of 5%), and significance was determined by Welch's T-test. **e** Defined macrophage populations isolated from harvested PDX tissue post-

carboplatin or control treatment, and their relative expression of M1 phenotypic markers. **f** Quantification and relative proportion of macrophage upregulation of *Ccl5* and *Cxcl10* genetic elements encoding chemoattractants within carboplatin-treated ($n = 245$ cells) and vehicle control tissue ($n = 135$ cells). **g** UMAP characterizing distinct populations of CAFs present in harvested tissue. **h** Dot plots showing normalized gene expression of ECM-associated constituents, immunosuppressive agents and chemotactic factors by cancer-associated fibroblasts (CAFs). **i** Violin plots showing the proportion of carboplatin-treated ($n = 19$ cells) or control-treated ($n = 73$ cells) endothelial cell markers associated with the tumor-associated high endothelial venule (TA-HEV) cellular profile responsible for regulation of lympho-cyte trafficking. Significance was determined by GSEA statistical and empirical evaluation to yield a normalized enrichment score (NES; significance level of 5%). Box plots in **d**, **f**, and **I** show the first to third quartile with median, whiskers show the minimum and maximum. Source data are provided as a Source Data file.

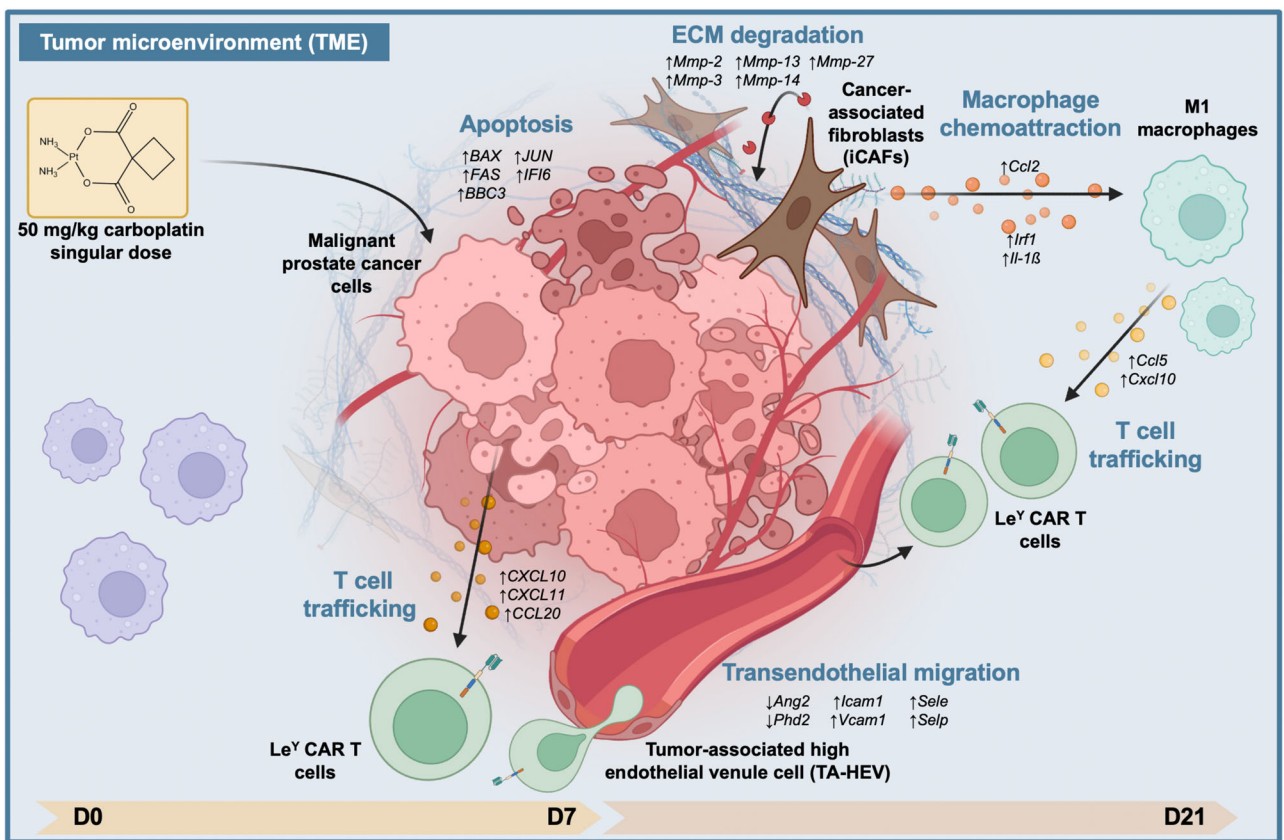

**Fig. 7 | Schematic defining the mechanistic actions of carboplatin on the tumor microenvironment of PDX grafts.** Carboplatin induced a series of alterations to the tumor microenvironment (TME) that collectively facilitate Le$^Y$ CAR T cell entry and persistence in vivo. These mechanisms include tumor cell apoptosis by day 7 (D7), followed by a cascade of changes to the TME by day 21 (D21), including ECM degradation, induction of an iCAF (inflammatory carcinoma-associated fibroblast) phenotype, CAF-secretion of macrophage chemoattractants, recruitment and polarization of M1 macrophages, macrophage-secretion of chemokines and cytokines that attract T cells, and an increase in tumor-associated high endothelial venule (TA-HEV) cells that enhance trans-endothelial cell migration and facilitate T cell trafficking into the tumor. This image was created with BioRender.com.

Glutamax, and 600 units/ml IL-2. The media was changed every two days until day 13.

In the Le$^Y$-CAR construct, a CD34 epitope was embedded for purification. During the CAR-T cell production, the transduced T cells were purified using the EasySep™ Human CD34 Positive Selection Kit (STEMCELL Technologies) on day 4–5 post transduction. The purified cells were further expanded for infusion. The percentage of CAR$^+$ cells in the final product was no <98%. The gating strategy for CAR T cells is shown in Supplementary Fig. 6a.

**In vitro CAR T cell killing assay**
Once organoids had reached approximately 50 µM in size, organoids were recovered from Matrigel using Cell Recovery Solution (Corning, USA #354253) for 1 h on ice. 50–100 organoids were plated in 10 µl of Matrigel to form a Matrigel dome in µ-Plate 96 well ibiTreat (ibidi GMBH, Germany #89626) with organoid media. In all, $3 \times 10^4$ Le$^Y$ CAR T cells or control ev-T cells were added to triplicate organoid wells with 300 units/ml IL-2. Live CAR T/ev-T cells can infiltrate into the dome, but the dead ones are excluded. Propidium iodide (PI, Sigma-Aldrich) was used as an indicator for cell death (2 µg/ml) and 1% Triton-X100 was used to obtain maximum cell death (positive control). Live cell images were captured by IX83 widefield live-imaging microscope (Olympus) over a 48 h period. Briefly, 10 fields were imaged per well at ×10 magnification at one time point and 25 time points were taken with 2 h interval. Images were analyzed using Fiji[45] and Bio-formats[46]. Approximately 50–100 organoids were analyzed individually per well and the MFI of PI of each organoid at each time point was measured.

Organoids were gated frame-by-frame to determine the PI uptake by dead organoid cells. For mechanism study, granzyme B inhibitor Compound 20 (10 µM) and perforin inhibitor SN34960 (20 µM) were used during the assay.

**Analysis of Le$^Y$ expression on tumor cell lines**
To analyze the Le$^Y$ expression on different prostate cancer cell lines, flow cytometry was used (Supplementary Fig. 6b). Cells were stained with Live/Dead Near-IR fixable viability dye (Thermo Fisher) for 10 min at 4 °C, then followed by a 30-min surface staining with a murine monoclonal antibody against human Le$^Y$ (clone 3S193) conjugated with Alex Fluor 488. Cells were washed and fixed in 2% paraformaldehyde then resuspended in FACS buffer before acquisition on a LSR Fortessa (Becton Dickinson). Flow cytometry data was then analyzed using FlowJo v10.8 software.

**$^{51}$Chromium release assay in tumor cell lines**
Target cells (Le$^{Y+}$ cell lines DU-145, PC-3, and 22Rv1 and Le$^{Y-}$ cell line MDA-MB435) were labeled with 50 µCi $^{51}$Chromium ($^{51}$Cr, PerkinElmer) for 1 h at 37 °C. The labeled cells were washed and subsequently cocultured with Le$^Y$-CAR-T cells or empty vector-transduced T cells in triplicate wells at effector to target ratios ranging from 50:1 to 6.25:1. Target cells alone (spontaneous release) and target cells with 10% Triton X-100 (maximum release) served as controls. After 4 h coculture, supernatants were collected, and the amount of $^{51}$Cr released (experimental release) was detected using a gamma counter (Wallac Wizard 1470, PerkinElmer). The % lysis was calculated by

[(experimental release – spontaneous release)/(maximum release – spontaneous release)] × 100.

## PDX experiments

All animal care and procedures were performed in accordance with Monash University animal ethics approval 20374 and Peter MacCallum Cancer Centre animal ethics approval E647. To assess the long-term in vivo response to Le$^Y$ CAR T cells, tissue from PDX-287R and PDX-224R-Cx was established subcutaneously (1 graft/mouse) in male NSG host mice (6–8 weeks old) until tumor volume reached approximately 50–100 mm$^3$. Mice received a single dose of irradiation (0.5 Gy) for lymphodepletion on day 0, before receiving three doses of $1 \times 10^7$ adoptively transferred Le$^Y$ CAR T cells or empty vector T cells in PBS by intravenous injection on days 1, 2, and 3. Mice also received eight doses of 50000 IU of IL-2 by intraperitoneal injection on days 1–5, 7, 9, and 11. A subset of host mice were pre-treated with 200 µg/dose nivolumab (1 dose every 5 days by intraperitoneal injection starting 5 days before day 1 of CAR T cell transfer, 4 doses in total, dissolved in PBS), 10 mg/kg docetaxel (1 dose 7 days before day 1 of CAR T cell transfer, dissolved in 90% PBS with 5% Tween 80 and 5% ethanol), or 50 mg/kg carboplatin (1 dose 7 days before day 1 of CAR T cell transfer, dissolved in water). The dose and dosing regimen of nivolumab has been used previously in NSG mice and is an equivalent safe dose for humans[47,48]. The doses of carboplatin and docetaxel are lower doses than in clinical practice to investigate the modulatory effects of these agents. IL-2 was purchased from Peprotech (Cranbury, New Jersey, USA), and nivolumab, docetaxel and carboplatin were purchased from Selleck Chemicals (Houston, Texas, USA). Between 5–21 mice were present per treatment group across 1–3 independent experiments (see figure legend). Tumors were measured three times a week for up to 6 weeks post-adoptive transfer of CAR T cells using calipers. Tumor volume was determined from caliper measurements by length × width × height × 0.52, as previously described[49], and animals were culled when tumor volume reached the maximum ethical limit of 1000 mm$^3$. At tumor collection, animals were placed under isoflurane for blood collection by cardiac puncture and then humanely killed by cervical dislocation. The investigators involved in treating mice were not blinded to the treatment groups but were blinded to tumor measurement data. Harvested tumor tissue was formalin-fixed for histological analysis. For early time point experiments, mouse PDXs were established as described above. Mice were treated with a single dose of 50 mg/kg carboplatin 7 days before the CAR T cell infusion and were irradiated 1 day before infusion. CAR T cells were given on day 1 and day 2 with 50,000 IU/dose IL-2. On day 3, 48 h after the first CAR T cell infusion mice were placed under isoflurane for blood collection by cardiac puncture, and then humanely killed by cervical dislocation. Blood, spleen, and tumors were collected for further analysis. For mechanistic single-cell RNA-sequencing studies, mice were treated with a single dose of 50 mg/kg carboplatin or vehicle control and tumor tissues were harvested on day 21.

## Immunohistochemistry on PDX tissue

Immunohistochemistry for phospho-histone H3 was performed on three representative sections per graft using the Leica BOND-MAX$^{TM}$ automated system (Leica Biosystems, Mount Waverley, Victoria, Australia). The primary antibody was a rabbit antibody against phospho-histone H3 (Ser10) Antibody (#9701), Research Resource Identifier (RRID): AB_331535 (Cell Signaling Technology) used at 8 µg/ml. The number of phospho-histone H3-positive cells were counted using Aperio ImageScope Software (Leica Biosystems), with three sections analyzed per graft, and expressed as a percentage of the total number of cells counted.

Immunohistochemistry for human CD3, human CD8, and mouse F4/80 was performed manually. FFPE slides were dewaxed in xylene and antigens were retrieved at 125 °C for 10 min in sodium citrate pH6

buffer (for CD3), Tris EDTA pH9 buffer (for CD8), or Tris EDTA pH8 buffer (for F4/80). Endogenous peroxidases were inactivated with 3% H$_2$O$_2$ (Merck), and the slides were blocked in PKI blocking buffer (Akoya Biosciences). Primary CD3 antibody (6.7 µg/ml; clone SP7; Abcam #ab16669; RRID AB_443425), CD8 antibody (1:100; clone 4B11; Invitrogen #MA1-80231; RRID AB_929437) or F4/80 antibody (1: 100; polyclone, Abcam #100790; RRID AB_10675322) was incubated 30 min at room temperature, and the secondary anti-mouse ImmPress (Vector Laboratories, MP-7402) or anti-rabbit ImmPress (Vector Laboratories, MP-7401) was added for 30 min at RT, followed by Dako Liquid DAB for 15 min. The slides were then counter-stained by Jung Autostainer (Leica) and scanned using the Olympus VS120 microscope at 20× magnification.

## Flow cytometric analysis of CAR T cells

To analyze CAR T cells in blood and spleen in vivo, flow cytometry was used for absolute cell counting and cell phenotyping. Blood and smashed spleen were treated with ACK buffer to remove the red blood cells. Cells were stained with Live/Dead Near-IR fixable viability dye (Thermo Fisher) for 10 min at 4 °C, then followed by 30-min surface staining for CD3 (1:100; BD Biosciences #564001), CD4 (BioLegend #317438), CD8 (1:100; BD Biosciences # 612889), CAR (as indicated by Flag-tag, 1:200; BioLegend #637310), and CD137 (1:100; BioLegend #309814), CD25 (1:200; BioLegend #302629). Before analyzing the cells on Symphony (BD Biosciences), cells were mixed with CountBright Absolute Counting Beads (ThermoFisher) for absolute cell count according to the manufacture's instruction. Flow cytometry data was then analyzed using FlowJo v10.8 software. The gating strategy is shown in Supplementary Fig. 6a.

## Analysis of tumor-infiltrating T cells

To analyze CAR T cells infiltrating tumors by flow cytometry, mice were euthanized 48 h-post first infusion, and tumors were digested using RPMI-1640 (Gibco) supplemented with 1 mg/ml collagenase type IV (Sigma-Aldrich) and 0.02 mg/ml DNase (Sigma-Aldrich). Tumors were incubated for 1 h at 37 °C, rendered to single-cell suspension, and then filtered twice (through 70-µm filter). Cells were then analyzed by flow cytometry directly for phenotyping and absolute cell counting using CountBright Absolute Counting Beads (ThermoFisher). For analysis of cytokines, single-cell suspensions were incubated on anti-idiotype antibody (0.5 µg/ml) or anti-CD3 antibody (1:100; clone OKT3, 0.5 µg/ml) coated-plate overnight with 25 units/ml IL-2 and were treated with GolgiStop (BD Biosciences) for 6 h before intracellular staining for IL-2 (1:50; BioLegend #500348), TNF-α (1:100; BioLegend #502912) and IFN-γ (1:100; BioLegend #502546). The gating strategy is shown in Supplementary Fig. 6a.

## Analysis of PD-L1 on organoids

To investigate CAR T cell-induced PD-L1 expression on organoids, cells from PDX-287R organoids were prepared on Matrigel with organoid growth media in 24-well plate as described above. Le$^Y$ CAR T cells were co-cultured with organoids for 24 h, then the supernatant was collected as CAR T cell-conditioned media. The fresh organoids or prostate tumor line DU-145 cells (as positive control) were incubated with either conditioned media or fresh media containing 150 ng/ml human IFN-γ (PeproTech) for 24 h. The cells were trypsinized and resuspended in single-cell suspension for flow cytometry analysis using anti-PD-L1 antibody (1:100; BioLegend #393608; Supplementary Fig. 6c, d).

## Dissociation of PDXs for single-cell analysis

PDXs were harvested from host mice and cut into 2 × 2 mm pieces using a scalpel. Tumor pieces were digested in RPMI-1640, containing 0.65 U/ml Liberase TM (Roche) and 0.2 mg/ml DNase I (Roche), for 1 hour at 37 °C, following by lysis of red blood cells (RBCs) using RBC Lysis Buffer (Sigma) for 1 minute. Cells were then resuspended in PBS,

1 mM CaCl2, with 2% FBS and underwent negative selection for viable cells using the Easy Sep Dead Cell Removal kit (Miltenyi), according to the manufacturer's protocol. Viable cells were passed through a 30 μM cell strainer (Miltenyi) to remove cell clumps, and then counted using Trypan blue. Samples with cell viability >80% were resuspended in PBS with 2% BSA and proceeded to single-cell analysis.

### Flow cytometric analysis and fluorescence-activated cell sorting (FACS) of PDXs and organoids

**Analysis of Lewis Y expression on PDX and organoids.** PDX-287R organoids were prepared on Matrigel with organoid growth media in 24-well plate as described above. Organoids were treated with 10 μM carboplatin for 24 or 72 h in triplicate. PDX tissue or treated organoids cultures were digested into single-cell suspensions as above. Cells were incubated with pre-titrated anti-human Lewis Y-Alexa Fluorochrome 647 (1:50–1:100; murine IgG3 monoclonal antibody, close 3S193) and IgG3-APC isotype control antibodies. Cells were incubated on ice in the dark for 15 min, washed with FACS buffer, and centrifuged for 5 min at 200 g to collect cells. Cells were filtered into 5 ml round-bottom tubes with strainer cap (Falcon®) prior to analysis. 100 ng/ml of propidium iodide (Sigma-Aldrich) was used to exclude dead cells. Samples were analyzed on the BD® LSR II Flow Cytometer. At least $2 \times 10^3$ live events were recorded for each file for downstream analysis using FlowJo™ v10.8.1. The gating strategy is shown in Supplementary Fig. 6c.

**Analysis of PDXs following carboplatin treatment.** PDX-287R and PDX-224R mice were established in host mice before being treated with a single dose of 50 mg/kg carboplatin or vehicle control. The tumor was harvested 7 days after treatment and digested into single-cell suspensions as above. Digested cells were incubated with pre-titrated antibodies diluted in staining buffer (1 x PBS with 10% FCS and 5 mM EDTA). Cells were pre-blocked for 5 min on ice with purified rat anti-mouse CD16/CD32 (25 mg/ml; clone 2.4G2, Cat #553141, BD Biosciences, USA) to block murine Fc receptors. Cells were then stained with the following antibodies (all antibodies are purchased from BioLegend unless otherwise stated); anti-human EpCAM-PECy7 (1:200; clone 9C4), anti-human CD95-BUV395 (4 mg/ml; clone DX2, BD Biosciences), anti-mouse MHC I-A$^k$-PE (0.2 mg/ml; clone 10-3.6), anti-mouse F4/80-APC (1 mg/ml; clone BM8), anti-mouse CD45-APCCy7 (1 mg/ml; clone 30-F11), anti-mouse ICAM1-BV421 (2 mg/ml; clone YN1/1.7.4), anti-mouse CD31-BV510 (2 mg/ml; clone 390, BD Biosciences) or isotype controls; mouse IgG1k-BUV395 (4 mg/ml; clone X40, BD Biosciences), Rat IgG2bk-BV421 (2 mg/ml; clone RTK4530). Cells were incubated on ice in the dark for 15 min, washed with FACS buffer, and centrifuged for 5 min at 200 g to collect cells. Cells were filtered into 5 ml round-bottom tubes with strainer cap (Falcon®) prior to analysis. In all, 100 ng/ml of propidium iodide (Sigma-Aldrich) was used to exclude dead cells. Samples were analyzed on the BD® LSR II Flow Cytometer. At least $2 \times 10^3$ live events were recorded for each file for downstream analysis with FlowJo™ v10.8.1.

For FACS, samples were sorted on the BD FACSAria™ Fusion Flow Cytometer at no more than $5 \times 10^3$ cells per second through a 100-micron nozzle at 20 psi. The following gating strategy was used to purified cell subsets; debris was excluded via FSC and SSC gate, PI was used to excluded dead cells and single cells were gated based on FSC-H and FSC-A parameters (Supplementary Fig. 6e). FACS purification of; tumor cells (EpCAM$^+$ CD45$^-$), immune cells (EpCAM$^-$ CD45$^+$), stromal cells (EpCAM$^-$, CD45$^-$, CD31$^-$) and endothelial cells (EpCAM$^-$, CD45$^-$, CD31$^+$). Samples were collected in 30% (v/v) FCS in RPMI, recovered by centrifugation, counted and analyzed for purity. Populations were sorted to greater than 90% purity.

### RNA isolation

Total RNA from FACS purified subsets were isolated using the RNA-queous Total RNA Isolation Kit (Ambion) according to the manufacturer's instruction. Total RNA was quantified in a Nanodrop ND-1000 spectrophotometer, checked for purity and integrity in a Bioanalyzer-2100 device (Agilent Technologies).

### Bulk RNAseq analysis

**Multiplex RNA-sequencing.** Samples with a RIN (RNA Integrity Number) $\geq$ 6, as determined by the Bioanalyzer-2100, were incorporated into the RNA library for associated downstream Multiplex RNA-sequencing and analysis. The 3' end of poly(A) primed transcripts were initially hybridized with a custom primer, 8 bp sample index and 10 bp unique molecular identifier (UMI). Subsequently, during first complementary DNA (cDNA) strand synthesis, a modified reverse transcription reaction facilitated the addition of a template switching sequence to the 5' end of the RNA transcripts. The amplified cDNA libraries were subsequently tagmented utilizing PCR transposase. Indexed complementary DNA (cDNA) was selected for and amplified utilizing Illumina P5 (5' AAT GAT ACG GCG ACC ACC GA 3') and P7 (5' CAA GCA GAA GAC GGC ATA CGA GAT 3')[50]. Multiplex RNA-sequencing was performed on the Illumina NextSeq550 by the Medical Genomics Facility (Monash Health Translation Precinct).

**Multiplex RNA-sequencing analysis.** The raw Fastq files were quality checked using FastQC and low-quality reads were trimmed using Cutadapt v1.7.1. Trimmed reads were aligned to both human hg38 and mouse mm39 reference genomes using STAR aligner v2.7.5b. Xeno-filteR v1.6 was used to select Human and mouse specific reads and counts matrix were generated using HTSeq v0.11.2. EdgeR v3.28 was used for differential expression analysis. The normalized log-transformed counts per million (CPM) was used to calculate single-sample gene set enrichment analysis score against specific MsigDB signature gene sets using R v4.2.0 package. Differentially expressed genes between treated vs untreated and its significance (adjusted P-value < 0.05) were calculated using Wald's T-test method implemented in the DESeq2 R package.

### Single-cell RNA transcriptome analysis (scRNA-seq)

**Single-cell RNA-sequencing.** scRNA-Seq for dissociated PDXs was performed using the 10X Genomics Chromium Single Cell 3' Library & Gel bead Kit V3.0, according to the manufacturers protocol (CG000183 Rev C). Briefly, ~5,000 PDX cells were used as input per sample. Cell encapsulation in microfluidic droplets yielded ~4000 recovered single-cell transcriptomes per sample. After reverse transcription, barcoded-cDNA was purified using SILANE Dynabeads followed by 11 cycles of PCR-amplification. SPRIselect purification was performed on an Agilent Bioanalyzer High Sensitivity chip to quantitate the fragment size and concentration of the amplified cDNA. Libraries were sequenced on an Illumina NovaSeq6000 with 151 bp paired-end reads.

**Single-cell RNA-Seq analysis.** XenoCell v1.0 was used to align transcripts to the GRCh38 human reference genome and mm10 mouse genome[51], mouse-specific cellular barcodes were selected containing a minimum of 90% of host-specific reads. Extracted mouse cells were then processed using Alevin tool (Salmon Software v1.3.0) to obtain unique molecular identifiers (UMIs; Supplementary Data 8) and generate a cell by gene count matrix[52], which was imported into Seurat (v3.2.0) for downstream analysis[53].

We excluded outlier cells that expressed in the range of <200–800 genes depending on the sample type and had unusual gene count, transcript count, and mitochondrial gene fraction, according to sample-specific thresholds (Supplementary Data 8). We also excluded genes expressed in fewer than 50 cells. Epithelial tumor cells (vehicle sample) identified were further down sampled to 100 cells, similar to the amount of carboplatin sample for all downstream analysis. For the carboplatin-treated sample, we subsampled to 500 cells, similar to the

amount of the vehicle sample. The SCTransform function from Seurat was then used to log-normalize and scale counts to 10,000 transcripts per cell and detect highly variable features. Principal component analysis (PCA) was performed using the top 3000 most highly variable features and used to cluster cells by Uniform Manifold Approximation and Projection (UMAP). Cell clusters were identified using Seurat FindClusters function. The ClusterTree (v0.4.3) R package was used to determine the optimal resolution and number of clusters for each sample[54]. Expression of selected genes signatures were calculated per-cell using the AddModuleScore function from Seurat. Differentially expressed markers were identified using Seurat FindMarkers function and significant genes were selected based on the FDR < 0.05. Gene set enrichment analysis was performed using escape (v1.2.0) enrichIT function in R (v4.1.0)[55].

Cell type annotation was performed by a differential gene expression analysis with a ROC test using FindAllMarkers function. Markers were ranked based on Log2Fold change and the ROC power score. Cross reference of the markers and literature was done to annotate each cell type[25].

Previously identified gene expression markers were used to define the phenotypes of cells present in the PDX graft. These signature genes were discerned by cross-examining multiple primary research papers detailing markers of M1/M2 polarization[26–28], the distinctions between an iCAF and myCAF phenotype[25,29], and specialization of the endothelium[30].

### Statistical analysis
Linear mixed model analyses were conducted using SPSS Statistics (Version 27; International Business Machines Corporation, Armonk, New York, United States of America). All other statistical analyses were performed in Prism v9 software (GraphPad). A $p$-value $\leq 0.05$ was considered statically significant. Unless otherwise state, data analyses were conducted using unpaired Student's $t$ test to compare two data sets or using one-way/two-way ANOVA when analyzing multiple sets of data. Data were presented as mean ± standard error of the mean (SEM). Statistically significant (FDR < 0.05) gene sets were calculated using escape package (v1.2.0) in R (v4.1.0).

### Reporting summary
Further information on research design is available in the Nature Portfolio Reporting Summary linked to this article.

## Data availability
The bulk RNA-sequencing, and single-cell RNA-sequencing data that support the findings of this study have been deposited in the NCBI dbGaP repository under accession number: phs003369.v1.p1. Genomic alterations in PDXs, determined by targeted DNA sequencing, was previously published. The curated set of genomic alterations was based on data downloaded from the cBioPortal resource Prostate Adenocarcinoma [https://www.cbioportal.org/study/summary?id=prad_p1000 and https://www.cbioportal.org/study/summary?id=prad_su2c_2015]. The remaining data are available within the Article or its Supplementary Information. Source data are available as Source Data File. To request access to MURAL PDXs and/or biospecimens, researchers should contact Dr. Melissa Papargiris, MURAL Project Manager (melissa.papargiris@monash.edu) to initiate an Expression of Interest. Researchers would need to provide evidence of institutional approval to experiment with human PDX tumors and research would be conducted under the conditions of a Materials Transfer Agreement. Source data are provided with this paper.

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

## Acknowledgements

We also acknowledge the members of the Prostate Cancer Research program, the patients, families, and consumers who support our research, and the members of the Melbourne Urological Research Alliance (MURAL). We wish to thank the Prostate Cancer Biorepository Network (PCBN) for access to the tissue microarray, whose work is supported by the Department of Defense Prostate Cancer Research Program, DOD Award No W81XWH-18-2-0013, W81XWH-18-2-0015, W81XWH-18-2-0016, W81XWH-18-2-0017, W81XWH-18-2-0018, and W81XWH-18-2-0019 PCRP PCBN. We acknowledge the association with the Bristol-Myers Squibb (BMS) study #CA209-7TK and thank BMS for providing access to the Le$^Y$ CAR T cell construct. We also thank Andrew Scott (Oliver Newton-John Cancer Research Institute) for access to the anti-Le$^Y$ antibody. This work was supported by Movember and MRFF (Upfront PSMA Prostate Cancer Research Alliance). Fellowship and grant support was obtained from the National Health and Medical Research Council (GPR, APP1102752: PKD, APP1136680; JAT, APP1102752; RAT, LHP, Ideas grant, APP2011391), Cancer Council Victoria (JAT, LHP, TP834128), Department of Health and Human Services acting through the Victorian Cancer Agency (RAT; MCRF15023), Prostate Cancer Foundation of Australia (LHP, PIRA YI-0322); and Monash University (LHP, Bridging Fellowship; LHP, Platform Access Grant PAG2-9067770177).

Infrastructure support was obtained from the EJ Whitten Foundation, Movember Foundation (Global Action Plan 1), the Peter and Lyndy White Foundation and TissuPath Pathology. This research was supported by the Monash University Histology Platform, Monash University Animal Research Laboratories, Monash Biomedicine Discovery Institute Organoid Program. This research was funded by a Prostate Cancer Research Alliance (PCRA) funded by Movember and the Medical Research Future Fund (MRFF). ProSTIC is additionally supported by the Prostate Cancer Foundation (PCF).

## Author contributions

G.P.R. and R.A.T. had full access to all the data in the study and takes responsibility for the integrity of the data and the accuracy of the data analysis. Concept and design: G.P.R., R.A.T., J.A.T., P.K.D., and P.J.N. Acquisition of data: L.H.P., J.J.Z., N.L.L., S.G.H., S.K., D.L.G., R.Q.U., and D.J.B. Analysis and interpretation of data: L.H.P., J.J.Z., N.L.L., S.G.H., S.K., D.L.G., R.Q.U., D.J.B., G.P.R., R.A.T., J.A.T., P.K.D., and P.J.N. Drafting of the manuscript: L.H.P., J.J.Z., N.L.L., S.G.H., S.K., R.Q.U., G.P.R., R.A.T., J.A.T., P.K.D., and P.J.N. Critical revision of the manuscript for important intellectual content: L.H.P., J.J.Z., N.L.L., S.G.H., D.J.B., A.A., I.V., M.S.H, P.J.N., P.K.D., J.A.T., R.A.T., and G.P.R. Statistical analysis: L.H.P., J.J.Z., N.L.L., S.G.H., S.K., and R.Q.U. Obtaining funding: G.P.R., R.A.T., J.A.T., and L.H.P. Supervision: G.P.R., R.A.T., J.A.T., P.K.D., and P.J.N. Other: none.

## Competing interests

G.P.R. and R.A.T. (Research collaborations: Pfizer, Astellas, Zenith Epigenetics, AstraZeneca); P.K.D. (Research funding from Myeloid Therapeutics, Prescient Therapeutics, and Bristol Myers Squibb); A.A. (Speakers Bureau: Astellas, Janssen, Novartis, Amgen, Ipsen, Bristol Myers Squibb; Merck Serono, Bayer; Honoraria: Astellas, Novartis, Sanofi, AstraZeneca, Tolmar, Telix, Merck Serono, Janssen, Bristol Myers Squibb, Ipsen, Bayer, Pfizer, Amgen, Noxopharm, Merck Sharpe Dome; Scientific Advisory Board: Astellas, Novartis, Sanofi, AstraZeneca, Tolmar, Pfizer, Telix, Merck Serono, Janssen, Bristol Myers Squibb, Ipsen, Bayer, Merck Sharpe Dome, Amgen, Noxopharm; Travel + Accommodation: Astellas, Merck Serono, Amgen, Novartis, Janssen, Tolmar, Pfizer; Investigator Research Funding: Astellas, Merck Serono, AstraZeneca; Institutional Research Funding: Bristol Myers Squibb, AstraZeneca, Aptevo Therapeutics, Glaxo Smith Kline, Pfizer, MedImmune, Astellas, SYNthorx, Bionomics, Sanofi Aventis, Novartis, Ipsen); M.S.H. (personal fees for lectures or advisory boards: Jannsen, Mundipharma, Astellas, Merck/MSD, Astra Zeneca, Point Biopharma; Research support paid to institution: Endocyte and Advanced Accelerator Applications, both Novartis companies); I.V. (Honoraria: Astellas, Abbvie, Tolmar, Janssen; Scientific Advisory Board: Astellas, AstraZeneca, Janssen, Bayer); P.J.N. (Research funding from BMS, Roche Genentech, MSD, Prescient Therapeutics, CRISPR Therapeutics, Allergan, Compugen); All other authors declare no competing interests.

## Additional information

[1]Prostate Cancer Research Group, Monash Biomedicine Discovery Institute, Cancer Program, Department of Anatomy and Developmental Biology, Monash University, Clayton, VIC 3800, Australia. [2]Cancer Immunology Program, Cancer Research Division, Peter MacCallum Cancer Centre, Melbourne, VIC 3000, Australia. [3]Sir Peter MacCallum Department of Oncology, The University of Melbourne, Parkville, VIC 3010, Australia. [4]Prostate Cancer Research Group, Monash Biomedicine Discovery Institute, Cancer Program, Department of Physiology, Monash University, Clayton, VIC 3800, Australia. [5]Cancer Research Division, Peter MacCallum Cancer Centre, Melbourne, VIC 3000, Australia. [6]Computational Cancer Biology Program, Peter MacCallum Cancer Centre, Melbourne, VIC 3000, Australia. [7]Department of Pathology, Peter MacCallum Cancer Centre, Melbourne, Victoria, Australia. [8]Department of Medical Oncology, Peter MacCallum Cancer Centre, Melbourne, VIC 3000, Australia. [9]Queensland Bladder Cancer Initiative, School of Biomedical Science, Faculty of Health, Queensland University of Technology, Brisbane, QLD 4102, Australia. [10]Australian Prostate Cancer Research Center, School of Biomedical Science, Faculty of Health, Queensland University of Technology, Brisbane, QLD 4102, Australia. [11]Department of Urology, Princess Alexandra Hospital, Brisbane, QLD 4102, Australia. [12]Molecular Imaging and Therapeutic Nuclear Medicine, Peter MacCallum Cancer Centre, Melbourne, VIC 3000, Australia. [13]Prostate Cancer Theranostics and Imaging Centre of Excellence (ProSTIC), Peter MacCallum Cancer Centre, Melbourne, VIC 3000, Australia. [14]These authors contributed equally: L. H. Porter, J. J. Zhu. [15]These authors jointly supervised this work: R. A. Taylor, G. P. Risbridger. ✉e-mail: renea.taylor@monash.edu; gail.risbridger@monash.edu

