## [Peer Review File · Nature Communications]

REVIEWER COMMENTS

Reviewer #1 (expertise in prostate cancer, urology):

In this study, the authors claimed that Low-dose carboplatin could remodel the tumor microenvironment to augment LeY CAR T cell efficacy in established human prostate tumors. LeY was shown to be expressed in a portion of prostate cancer (PCa) specimens. Therefore, it is reasonable to use LeY CAR T for LeY+ PCa treatment. Also, since LeY expression seemed to be irrelevant to the AR status, the authors claimed that LeY CAR T should be able to target AR+ adenocarcinoma as well as AR- NEPC.

PCa is a well-known immune "cold" tumor, and indeed CAR T therapy alone did not reduce tumor regression in their PDX-287R mouse model. Therefore, the authors tried to use combinational chemotherapy to remodel the TME to improve LeY CAR T infiltration and function. The authors found that carboplatin could significantly remodel the TME by altering the cancer-associated fibroblast phenotype, enhancing extracellular matrix degradation, and attracting macrophages and re-orienting their M1 polarization. As a result, low-dose carboplatin treatment enhanced CAR T trafficking and retention in the tumor mass and caused PDX-287R tumor regression. Overall, the manuscript is very well written and the results are solid. However, this study lacks some novelty and needs some improvement in the experimental design.

Major comments:

#1, The authors used one PDX model (PDX-287R) for in vivo study. PDX-287R happens to be extremely sensitive to carboplatin treatment while resistant to docetaxel treatment. Will docetaxel cause similar TME changes in docetaxel-sensitive models? To rule out the possibility that it is an outlier effect, it would be helpful to include more models with various carboplatin-sensitivity and docetaxel-sensitivity.

Cytotoxic chemo drugs have been found to be able to induce immunogenic cell death and dampen suppressive TME. In this paper, the authors only used one PDX model, i.e. PDX-287R, and found carboplatin, but not docetaxel, caused TME remodeling and tumor regression. However, docetaxel is an effective chemo drug for PCa treatment. It has been reported that docetaxel can remodel PCa immune microenvironment and enhance checkpoint inhibitor-based immunotherapy or PMSA CAR T therapy (PMID: 35836810; PMID: 32728611). Also, the combination of docetaxel plus immunotherapy has been used in multiple clinical trials for treating advanced PCa (NCT02861573, NCT03338790, etc). So far, the combination has been well tolerated and improved the overall response rates in some clinical studies.

PDX-287R model seems extremely sensitive to carboplatin treatment. Significant tumor death happened and residual tumor cells only accounted for 2.1% of the tumor mass 3 weeks post-carboplatin treatment (Fig 5a,b). Most of the tumor mass was occupied by mouse stroma cells, among them > 50% cells were monocytes and macrophages. Therefore, PDX-287R model may represent a docetaxel-resistant, carboplatin-sensitive tumor model, and carboplatin induced dramatic cell death and inflammation in this model. The TME remodeling described in this study may also happen after other effective chemo drug treatment.

#2, The authors suggest that compared to PMSA CAR T, LeY CAR T is more suitable for AR- NEPC treatment. If PDX-287R represents an AR+PSMA+NE- tumor, then it is worth studying the TME remodeling effect in some AR-PSMA-NE+ PDX tumor models listed in figure 1c.

Minor comments:

#1, What's the genomic background of PDX-287R? Any alterations of DNA damage repair genes that may cause increased sensitivity to carboplatin treatment.

#2, In figure 3, the authors compared the tumor growth of PDX-287R following various treatments. To confirm the LeY-specific killing in this in vivo model, carboplatin + control Car T (ev-T) treatment should also be included for comparison.

#3, The overall hypothesis is that carboplatin changes TME, induces macrophage and endothelium change, then recruits more T cells. So, examining the TME change at an earlier time point right before CAR T cell injection may be helpful to explain the contribution of TME change to T cell infiltration.

#4, Figure 1C shows that there are about 70% LeY positive cells in PDX-287R. However, LeY CAR T cell treatment results in tumor regression to <1% of the starting volume. How to explain the cell killing of LeY negative cancer cells? What is the expression level of LeY in the residual PDX-287R tumors after treatment? The data from models with ~100% LeY positive cells (e.g. 224R, 422M) and ones with ~0% LeY positive cells (e.g. 330M and 27.1A) will provide more evidence supporting the effects of combination treatment.

#5, Figure 3c, the starting tumor volume (tumor volume <10mm³) of docetaxel treatment seems to be much smaller than those with other treatments (tumor volume ~100mm³; Fig 3a,b,d).

#6, To show the success of CAR-T cell production and on-target action of LeY CAR T cells, authors used ev-T as control. It will be nice to also test CAR T mediated cytotoxicity on LeY negative or LeY knockout organoids.

#7, It will be helpful to provide IHC images (low power) of other PDXs and organoids (435.1A, 201.1A) to show the percentage of positive cells.

Reviewer #2 (expertise in CAR T cells, immunotherapy, prostate cancer):

In this manuscript "Low-dose carboplatin remodels the tumor microenvironment to augment CAR T cell efficacy in established human prostate tumors", the authors have demonstrated that the efficacy of CAR T cells directed against solid Le^Y-expressing prostate tumors can be considerably enhanced by carboplatin chemotherapy in vivo (in contrast to Le^Y-CAR T cell immunotherapy alone or in combination with docetaxel or nivolumab). To explain this, the authors have shown that carboplatin modulates the TME to overcome the immunosuppressive milieu. Mechanistically, carboplatin treatment of PDX-derived tumors in immunodeficient mice facilitates the infiltration of human CAR T cells into the tumors, results in an enrichment and persistence of CAR T cells and enhances CAR T cell activation at the tumor site. Furthermore, this study shows that carboplatin treatment resulted in an accumulation of mouse macrophages showing pro-inflammatory M1 polarization and a shift of CAFs to a pro-inflammatory phenotype in the tumors. Generally, this work provides validated noteworthy results and novel significant findings to the field. The methodology is sound and detailed explained. There are only a few minor remarks/suggestions/questions to this manuscript for the authors.

The study shows that Le^Y is expressed at different levels on human prostate cancer, in PDXs and organoids established from PDXs. Please define percentage/range for low, moderate and high Le^Y expression mentioned in Figure 1 and 2 more precisely.

The authors have demonstrated that Le^Y-directed CAR T cells specifically kill organoids established from PDXs in vitro. Please explain how do you distinguish between PI uptake by organoids or by (CAR-) T cells in your in vitro killing assay (Figure 2).

The authors have clearly shown that the killing effect in vitro correlates with the Le^Y expression level. Would be interesting to know whether you have measured a threshold/window of Le^Y expression level on tumor cells that is required for significant killing by Le^Y CAR T cells. Availability of the targeted antigen is required for CAR T cell activity and can be altered in vivo. Have you measured/confirmed the Le^Y expression on the tumors of PDX grafts in experimental mice? The authors have shown that carboplatin has an effect on several factors. Have you investigated whether carboplatin also changes the expression of Le^Y?

This study shows that carboplatin considerably enhances anti-tumor CAR T cell effect in vivo. Furthermore, carboplatin increases CAR T cell activation and cytokine production. In detail, you have measured the IL-2-, TNF- α - and IFN- γ -positive TILs using intracellular staining. Thus, the conclusion should be that carboplatin increases cytokine production rather than secretion (page 20, line 454-455), right? Would be of interest whether carboplatin also has an effect on CAR T cell proliferation.

The authors have shown an enrichment of mouse macrophages showing M1 polarization in carboplatin-treated PDX tumors in immunodeficient mice. Do you expect the same effect on human macrophages? This aspect should be considered/discussed.

Other minor comments:

Would be good to explain the treatment plan for nivolumab described in the Methods part in the section PDX experiment (page 10, line 232-234).

In the Methods part (page 12, line 271) a bracket is missing.

In Figure 1 legend: Please check whether tumors were considered positive with $>10\%$ or $\geq 10\%$ membrane Le^Y-positive cells. (In the text is written " $\geq 10\%$ membrane Le^Y-positive cells".)

In Figure 1: Please define "NE" markers in the figure legend.

In Supplementary Figure 1a: Please define "Iso" in grey in the figure legend.

In Supplementary Figure 1b: You have performed statistical analysis CAR-T versus media and IFN- γ versus CAR-T, correct? You should also include statistical analysis for IFN- γ versus media.

In Figure 3: Please change the labeling 3"E" into 3"e".

In Figure 4: Please explain the abbreviation "DTX" in the figure legend.

In the Results part describing the CAR T cell activation and cytokine measurements you refer to Fig.2i-j (page 20, line 456-457). Should this be Fig.4i-j?

Reviewer #3 (expertise in CAR T cells, T cell engineering):

Porter and colleagues demonstrated in an immunodeficient mouse model of prostate cancer, that the anti-tumor response of anti-Le^Y CAR T cells could be significantly enhanced by combination with carboplatin-based chemotherapy. In addition to direct cytotoxic effects on tumor cells, the authors demonstrate that carboplatin is able to remodel the TME and propose that these changes may lead to enhanced CAR-T cell infiltration and activation, ultimately culminating in enhanced tumor eradication.

As demonstrated by the low response rates of CAR-T cells in clinical trials, the need for new therapeutic strategies for solid tumor is high. Therefore, the presented results could guide the development of new combination therapies for prostate cancer.

However, before publication there are still some remaining questions that should be addressed by the authors.

Major:

1. Figure 5: Authors demonstrate that monotherapy with carboplatin alone results in an effective elimination of tumor cells 3 weeks after therapy (2.1% tumor cells), underlining that carboplatin-therapy alone is already very efficient in eliminating tumor cells. However, according to Figure 3D, carboplatin has only a minor effect on tumor size. How do authors explain these discrepancies? Are tumors measured in Figure 3D mainly consisting of TME rather than tumor cells? If so, which additional effects do Le_Y CAR-T cells exert? Is there a cross-reactivity of the anti-Le_Y binding domain of CARs with murine Le_Y expressed e.g. on murine TME? Please provide in addition to Figure 5A, a diagram showing tumor volume changes in vehicle and carboplatin group over time.

Since most tumor cells were eliminated after 3 weeks, TME changes should be monitored at earlier time points. The CAR-T group should also be included.

2. The authors have extensively studied carboplatin-induced changes within the TME. However, to obtain a complete picture of the underlying mechanisms of how carboplatin enhances the anti-

tumor response of Le_Y CAR-T cells, the authors should also examine carboplatin-induced alterations on tumor cells: Do tumor cells change protein expression levels of Le_Y antigen upon carboplatin treatment? Do cancer cells alter the expression of co-stimulatory (e.g. 4-1BBL, Ox-40L, CD80, CD86) or co-inhibitory molecules (e.g. PD-L1, PD-L2, ...)? Do such carboplatin-induced changes on tumor cells, enhance the antitumor activity of CAR-T cells?

3. To analyze the effect of carboplatin on the TME, the authors chose an immunodeficient mouse model in which most immune cells are absent. Normally, TME studies are performed in syngeneic models or other immunocompetent mouse models. The authors should discuss the advantages and disadvantages of their model. How might the results observed in immunodeficient mice translate to humans?

Minor:

4. The authors should comment on the chosen dosing regimen and the applied concentrations of cisplatin, carboplatin and nivolumab in their combination studies with CAR-T cells. Are the applied concentrations clinically relevant/achievable. Were the drugs used similarly to clinical settings?

5. Immunohistochemistry was used to analyze Le_Y expression in human prostate cancer samples. Based on which parameters were tumors classified as "low", "moderate" and "high"? What was the cut-off?

6. What was the transduction rate of CAR-T cells infused into the mice for treatment? If the transduction rate was not 100%, CD3 or CD8 staining cannot be used to measure infiltration of CAR-T cells. In this case, direct staining of CAR-T cells must be performed.

Response to reviewers:

We thank all reviewers for their insightful comments and recommendations.

In our revised manuscript, we provide a comprehensive set of new experiments, including complex *in vivo* PDX studies. We believe that the new data improve the veracity of our work, and supports the conclusions of our study, providing compelling preclinical evidence for the use of carboplatin as a modulating agent for CAR T cell immunotherapy in the treatment of prostate cancer.

Our study is novel for several reasons:

- There are still no approved CAR T cell therapies for prostate cancer, despite significant effort in the field.
- The exact therapeutic strategies to overcome the immunosuppressive TME and maximise CAR T cell efficacy remain undefined, yet is essential for effective future clinical trial design.
- We report Le^Y as an unexpected tumor-associated antigen for prostate cancer, with utility in a broad range of tumor types, not only PSMA- or PSCA- positive tumors.
- Both therapies (Le^Y CAR T cells and carboplatin) are approved for use in patients and could be tested in clinical trials immediately, based on these preclinical observations.

In our rebuttal, we have provided **substantial new data, with 17 new panels in the main figures and 29 new panels in supplementary data**. These include:

- Analysis of a second PDX line, PDX-224R Cx, which showed reduced sensitivity to carboplatin treatment alone and decreased CAR T cell infiltration; however, a reduction in tumor burden with increased T cell activation was still observed (involved 9 weeks graft establishment time prior to treatment and 7 weeks follow-up post treatment).
- Mechanistic studies showing the difference in response between the two PDX lines, PDX-287R and PDX-224 Cx (involved treatment of grafts *in vivo*, with 6-9 weeks graft establishment time prior to treatment).
- Changes within the TME at one week post carboplatin treatment, the time of CAR T cell infusion, using flow cytometry and RNA sequencing.
- Additional *in vivo* experiments with the carboplatin + empty vector T cell control group for PDX-287R (involved 6 weeks graft establishment time prior to treatment and 7 weeks follow-up post treatment).
- Additional experiments confirming on target action of Le^Y CAR T cells *in vitro* using Le^Y-negative and Le^Y-positive cell lines.

Please find below a point-by-point response to individual review comments.

Reviewer #1 (expertise in prostate cancer, urology)

Major comments:

1. The authors used one PDX model (PDX-287R) for *in vivo* study. PDX-287R happens to be extremely sensitive to carboplatin treatment while resistant to docetaxel treatment. Will docetaxel cause similar TME changes in docetaxel-sensitive models? To rule out the possibility that it is an outlier effect, it would be helpful to include more models with various carboplatin-sensitivity and docetaxel-sensitivity.

Cytotoxic chemo drugs have been found to be able to induce immunogenic cell death and dampen suppressive TME. In this paper, the authors only used one PDX model, i.e. PDX-287R, and found carboplatin, but not docetaxel, caused TME remodeling and tumor regression. However, docetaxel is an effective chemo drug for PCa treatment. It has been reported that docetaxel can remodel PCa immune microenvironment and enhance checkpoint inhibitor-based immunotherapy or PMSA CAR T therapy (PMID: 35836810; PMID: 32728611). Also, the combination of docetaxel plus immunotherapy has been used in multiple clinical trials for treating advanced PCa (NCT02861573, NCT03338790, etc). So far, the combination has been well tolerated and improved the overall response rates in some clinical studies.

PDX-287R model seems extremely sensitive to carboplatin treatment. Significant tumor death happened and residual tumor cells only accounted for 2.1% of the tumor mass 3 weeks post-carboplatin treatment (Fig 5a,b). Most of the tumor mass was occupied by mouse stroma cells, among them > 50% cells were monocytes and macrophages. Therefore, PDX-287R model may represent a docetaxel-resistant, carboplatin-sensitive tumor model, and carboplatin induced dramatic cell death and inflammation in this model. The TME remodeling described in this study may also happen after other effective chemo drug treatment.

Response: We thank the reviewer for these thoughtful comments. We agree that the initial PDX model, PDX-287R, showed a significant response to carboplatin, resulting in cell death and inflammation that augmented CAR T cell efficacy. Whilst sensitivity to carboplatin is a clearly a key feature, the transcriptomic changes following carboplatin treatment strongly suggest that the pro-inflammatory effects are not solely dependent on cancer cell death. These changes suggest that tumor stromal cells are also impacted by carboplatin, leading to the secretion of pro-inflammatory cytokines and chemokines. These data support the conclusion that if the appropriate TME modifications are induced, the CAR T cells can access the tumor and become activated to enact their killing function. Immunogenic cell death in this model is made less likely by the fact that the host lacks a lymphoid cell compartment to initiate a cognate cell response. As these studies were conducted in NSG mice with no T cells, the mechanisms of immunogenic cell death cannot solely account for the effects that we saw. Therefore, TME changes at the transcriptional level likely play a major role in CAR T cell infiltration and tumour response, which is a highly novel finding.

To further support this conclusion, we have looked at the response of PDX-287R to carboplatin and docetaxel at one week post treatment, the time of T cell infusion. Carboplatin induced tumor cell apoptosis and myeloid cell infiltration at this time; however, apoptosis and myeloid cell infiltration was not increased at this time after docetaxel treatment.

This notion was supported by our additional studies in another PDX, PDX-224R-Cx, where the response to carboplatin was modest and the infiltration of CAR T cells into the tumor was greatly reduced. Therefore, carboplatin improves the efficacy of CAR T cell treatment, with the extent of the response dependent on changes induced within the tumor. Again, this is not only dependent on the effect of cancer cells alone as apoptosis was not increased at the time of T cell infusion; however, we still saw increased CAR T cell activation within the grafts and a reduction in tumour burden. Taken together, these data suggest that a pro-inflammatory gene signature induced by agents such as carboplatin may be helpful in identifying patients that may benefit from CAR T cell therapy. We do agree with the referee that further work is required to precisely define the biomarkers that will aid patient selection in the clinic.

As for a docetaxel-sensitive model, we have tested the response to this chemotherapy in 18 different PDXs in our collection, and none have shown a significant response *in vivo*. This is likely because our collection come from patients with high-risk, therapy resistant disease

where the tumors are refractory to standard-of-care agents. Therefore, we are unable to test the effect of combination therapy with docetaxel in a docetaxel-sensitive tumor. However, we agree there is evidence in the literature that is possible, and we have included this point in the discussion.

Revisions:

- The data for the new PDX line, PDX-224R-Cx is shown in **Fig. 3g** and **Supplementary Fig. 2j**, with corresponding text on page 23 (lines 537-543).
- Data showing CAR T cell infiltration and activation within this PDX tumor 48 hours post CAR T cell infusion is shown in **Fig. 4j-l** and **Supplementary Fig. 3e-h**, with corresponding text on page 25 (lines 569-575).
- Data showing the induction of tumor cell apoptosis and the proportion of cellular subsets by flow cytometry in PDX-224R-Cx grafts one week following carboplatin treatment compared to PDX-287R grafts is shown in **Fig. 5d-f** and **Supplementary Fig. 4f**, with corresponding text on page 25 (lines 582-585) and page 26 (lines 612-614).
- Additional data showing induction of apoptosis and the proportion of cellular subsets by flow cytometry in PDX-287R one week following docetaxel treatment is shown in **Supplementary Fig. 4g-l**, with corresponding text on page 26 (lines 614-616).
- We have modified the abstract (page 3, lines 70-74) and discussion (page 31, lines 714-718 and 727-731; page 32, lines 732-733) to incorporate this new data.
- We have included a paragraph in the discussion regarding the reports of docetaxel remodelling the immune microenvironment and enhancing checkpoint inhibitor-based immunotherapy or PMSA CAR T therapy in prostate cancer (page 33, lines 756-769).

2. The authors suggest that compared to PMSA CAR T, LeY CAR T is more suitable for AR-NEPC treatment. If PDX-287R represents an AR+PSMA+NE- tumor, then it is worth studying the TME remodeling effect in some AR-PSMA-NE+ PDX tumor models listed in figure 1c.

Response: The second PDX line included in the revised manuscript, PDX-224R Cx, is a neuroendocrine tumor that is AR-PSMA-. This line showed a partial response to the carboplatin-CAR T cell combination therapy. However, this is likely due to the extent of the response to carboplatin, and decreased CAR T cell infiltration into the tumor. A larger panel of AR+NE- and AR-NE+ lines is required to determine whether this response is phenotype dependent. However, as Le^Y is expressed on AR-NE⁺ tumors, and both adenocarcinoma and neuroendocrine tumors responded to Le⁺-CAR T cells *in vitro*, we still believe that targeting Le^Y may be an additional strategy for patients that would not benefit from AR- and PSMA-directed therapies, or who have progressed following these treatments.

Revisions:

- We have included a new paragraph in the discussion (page 35; lines 812-819) that indicates that a larger panel of AR+NE- and AR-NE+ lines is required to determine whether this response is phenotype dependent.
- We have removed the statement indicating Le^Y CAR T is more suitable for AR- NEPC treatment from the abstract and the concluding statement in the discussion.

Minor comments:

1. What's the genomic background of PDX-287R? Any alterations of DNA damage repair genes that may cause increased sensitivity to carboplatin treatment.

Response: We have included genomic alterations, based on targeted DNA sequencing, for all six PDX lines used in the organoid assays, including PDX-287R. These genomic alterations were previously reported for these PDXs (PMID: 34413304).

PDX-287R has a mutation in the mismatch repair gene mutS homologue 2 (*MSH2*), which has not been associated with sensitivity to carboplatin. However, it has been associated with improved long-term response to immune checkpoint inhibitors in a range of solid tumors (PMID: 31151482). This may have contributed to the response seen here and needs to be investigated in other prostate tumors with mismatch repair deficiency.

Revisions:

- Genomic alterations, based on target DNA sequencing, for PDX lines have been showed in **Supplementary Fig. 1c**, with corresponding text on page 21, line 490-491 of the manuscript.
- Discussion of the mismatch repair deficiency of PDX-287R has been included in the discussion (page 33, line 770-774).

2. In figure 3, the authors compared the tumor growth of PDX-287R following various treatments. To confirm the LeY-specific killing in this in vivo model, carboplatin + control Car T (ev-T) treatment should also be included for comparison.

Response: We have repeated the experiment to include the carboplatin + ev-T cell control group. Following treatment with carboplatin + ev-T cells, there was an initial decrease in tumor burden before tumors grew back, consistent with the response of grafts treated with carboplatin alone.

Revisions:

- The repeated experiment for PDX-287R with the carboplatin + ev-T cells control group is shown in **Supplementary Fig. 2d**, with corresponding text on page 23 (lines 521-524).
- Data showing carboplatin + evT cell treatment group for PDX-224R-Cx is shown in **Fig. 3g** and **Supplementary Fig. 2j**, with corresponding text on page 23 (lines 543-544) and page 24 (lines 545-546).

3. The overall hypothesis is that carboplatin changes TME, induces macrophage and endothelium change, then recruits more T cells. So, examining the TME change at an earlier time point right before CAR T cell injection may be helpful to explain the contribution of TME change to T cell infiltration.

Response: Thank you for this suggestion. We have now included data from flow cytometry and RNA sequencing at one week post carboplatin treatment, which is the time of CAR T cell infusion. For PDX-287R, we found that carboplatin induced a decrease in the proportion of EpCAM⁺ tumor cells and an increase in tumor cell apoptosis within the grafts. RNA sequencing revealed a shift towards a pro-inflammatory phenotype in the tumor cells, as well as increased cGAS-STING signalling and increased expression of chemokines, including CXCR3 ligands for T cell chemotaxis. This suggests that carboplatin induced a pro-inflammatory phenotype at the transcriptional level in PDX-287R tumor cells and could account for the increase T cell infiltration following carboplatin treatment.

At one-week post treatment, there was also an increase in the proportion of CD45⁺ immune cells in PDX-287R grafts. Despite these proportional changes, RNA sequencing revealed that there was not a shift towards the M1 pro-inflammatory macrophage phenotype at this

stage, as we previously reported for the three-week time point. This suggests that this shift in immune cell phenotype occurs downstream from the initial myeloid cell recruitment.

Revisions:

- Flow cytometry and RNA sequencing data for PDX-287R at the one-week time point has been included in a new figure, **Fig. 5**, with supplementary data in **Supplementary Fig. 4**.
- Corresponding text is on page 25 (lines 578-594), page 26 (lines 595-618) and page 27 (lines 619-620).

4. Figure 1C shows that there are about 70% Le^Y positive cells in PDX-287R. However, Le^Y CAR T cell treatment results in tumor regression to <1% of the starting volume. How to explain the cell killing of Le^Y negative cancer cells? What is the expression level of Le^Y in the residual PDX-287R tumors after treatment? The data from models with ~100% Le^Y positive cells (e.g. 224R, 422M) and ones with ~0% Le^Y positive cells (e.g. 330M and 27.1A) will provide more evidence supporting the effects of combination treatment.

Response: We have performed immunohistochemistry for Le^Y on treated grafts and found that Le^Y expression was low or absent in residual tumor cells following carboplatin and CAR T cell treatment. However, Le^Y expression was still high in tumors treated with carboplatin alone and vehicle control, suggesting that the lack of Le^Y expression in residual tumors was not due to carboplatin treatment but instead CAR-T cells targeting Le^Y-positive cells.

We also determined the expression of Le^Y in PDX-287R using flow cytometry as it is more sensitive than immunohistochemistry and found that 95% of tumor cells were positive for Le^Y expression. This could account for the loss of the majority of tumor cells within the grafts following Le^Y-CAR T cell treatment. Furthermore, this line was initially sensitive to carboplatin alone, which could have resulted in the death of Le^Y-negative tumor cells.

The second PDX line tested, PDX-224R Cx, had 90% positive Le^Y cells by immunohistochemistry. However, PDX-224R Cx showed only a partial response to treatment, demonstrating that pro-inflammatory changes induced by carboplatin are mandatory to enable the CAR T cells to gain unfettered access to the Le^Y-positive tumor cells and eliminate the tumor.

Revisions:

- Flow cytometric quantification of PDX-287R Le^Y expression is shown in **Supplementary Fig. 2a**, with corresponding text on page 22 (lines 510-512).
- The expression of Le^Y by immunohistochemistry in treated grafts is shown in **Supplementary Figure 2d-f**, with corresponding text on page 23 (lines 529-534).
- We have re-plotted tumor growth on a linear axis in a new supplementary figure (**Supplementary Fig. 5a**) to highlighted the response of PDX-287R to carboplatin treatment alone. See corresponding text on page 27 (lines 627-638).

5. Figure 3c, the starting tumor volume (tumor volume <10mm³) of docetaxel treatment seems to be much smaller than those with other treatments (tumor volume ~100mm³; Fig 3a,b,d).

Revision: Thank you for highlighting this error with the axis label. We have now re-graphed this to have the same axis scale as the carboplatin treatment groups in **Fig. 3c**.

6. To show the success of CAR-T cell production and on-target action of Le^Y CAR T cells,

authors used ev-T as control. It will be nice to also test CAR T mediated cytotoxicity on Le^Y negative or Le^Y knockout organoids.

Response: Unfortunately, we cannot generate organoids from all of our PDX models due to well-documented challenges with growing prostate cancer cells as organoids. To show the response to Le^Y-negative tumor cells, we used the human melanoma cell line MDA-MB435, demonstrating no CAR T cell mediated cytotoxicity compared to ev T cells. For comparison, we also included three Le^Y-positive prostate cancer cell lines, which showed increased lysis when co-cultured with increasing doses of Le^Y-CAR T cells.

Revision: The response of the Le^Y-negative tumor cell line, MDA-MB435, and the Le^Y-positive cell lines has been shown in a new supplementary figure (**Supplementary Fig. 1e**), with corresponding text on page 22 (lines 500-503) of the manuscript.

7. It will be helpful to provide IHC images (low power) of other PDXs and organoids (435.1A, 201.1A) to show the percentage of positive cells.

Revisions:

- Low power IHC images for Le^Y in the six PDXs and organoids used for the killing assays is shown in **Supplementary Fig. 1a**
- We have also graphed the percent of cells positive for membrane Le^Y staining in both the organoids and PDXs in **Supplementary Fig. 1b**
- We have added the percentage of cells positive for membrane Le^Y expression for each organoid on the main figure (**Fig. 2d**)
- The corresponding text for these changes is on page 22 (lines 508-512) of the manuscript.

Reviewer #2 (expertise in CAR T cells, immunotherapy, prostate cancer):

1. The study shows that Le^Y is expressed at different levels on human prostate cancer, in PDXs and organoids established from PDXs. Please define percentage/range for low, moderate and high Le^Y expression mentioned in Figure 1 and 2 more precisely.

Response: For Figure 1, we have included percentages for Le^Y expression in the figure legend, defining low as <10% (equivalent with the cut off into clinical trials), moderate as 10-50% and high as >50%. For Figure 2, we have added the percentage of cells positive for membrane Le^Y expression for each organoid on the main figure. In addition, to address Reviewer 1's comment above, we have a new Supplementary Figure with low power IHC images to show Le^Y expression in the six PDX lines and organoids used for the killing assays, and a graph to show the percentage of cells positive for membrane Le^Y expression in the PDXs and organoids.

Revisions:

- Amendments made to define low, moderate and high Le^Y expression included within **Fig. 1b** figure legend.
- Percentage of organoid cells positive for membranous Le^Y expression defined in **Fig. 2d** and **Supplementary Fig. 1b**, with corresponding text on page 22 (lines 498-500) of the manuscript.
- Immunohistochemical staining for Le^Y in PDXs and organoids is shown in **Supplementary Fig. 1a-b**.

2. The authors have demonstrated that Le^Y-directed CAR T cells specifically kill organoids

established from PDXs in vitro. Please explain how do you distinguish between PI uptake by organoids or by (CAR-) T cells in your in vitro killing assay (Figure 2).

Response: The organoids were seeded in a Matrigel dome and the CAR-T/ev-T cells were added subsequently outside the dome. Live CAR-T/ev-T cells can infiltrate into the dome, but the dead ones will be excluded. During the analysis, the organoids were also traced and gated frame-by-frame to measure the PI uptake by dead cells of the organoids.

Revision: Additional methodology including PI uptake by dead cells within Matrigel-seeded organoids provided on page 10 (lines 223-225) of the manuscript.

3. The authors have clearly shown that the killing effect in vitro correlates with the Le^Y expression level. Would be interesting to know whether you have measured a threshold/window of Le^Y expression level on tumor cells that is required for significant killing by Le^Y CAR T cells. Availability of the targeted antigen is required for CAR T cell activity and can be altered in vivo. Have you measured/confirmed the Le^Y expression on the tumors of PDX grafts in experimental mice? The authors have shown that carboplatin has an effect on several factors. Have you investigated whether carboplatin also changes the expression of Le^Y?

Response: We assessed Le^Y expression in treated PDX grafts by immunohistochemistry. Le^Y expression was low or absent in residual tumor cells following carboplatin treatment in combination with Le^Y CAR T cells, consistent with CAR-T cells targeting Le^Y-positive cells. However, Le^Y expression was still high in tumors treated with carboplatin alone and vehicle control, suggesting that carboplatin treatment alone does not alter Le^Y expression. To confirm this, we treated organoids derived from PDX-287R with carboplatin *in vitro* for 24 hours and 72 hours, and analyzed the expression of Le^Y by flow cytometry. This showed that carboplatin did not decrease Le^Y expression.

Revisions:

- Le^Y expression in treated grafts is showed in **Supplementary Fig. 2f-g**, with corresponding text on page 23 (lines 529-530).
- Le^Y expression in organoids following treatment with carboplatin *in vitro*, is showed in **Supplementary Fig. 2h-i**, with corresponding text on page 23 (lines 530-534).

4. This study shows that carboplatin considerably enhances anti-tumor CAR T cell effect in vivo. Furthermore, carboplatin increases CAR T cell activation and cytokine production. In detail, you have measured the IL-2-, TNF- α - and IFN- γ -positive TILs using intracellular staining. Thus, the conclusion should be that carboplatin increases cytokine production rather than secretion (page 20, line 454-455), right? Would be of interest whether carboplatin also has an effect on CAR T cell proliferation.

Response: We are interested to see whether carboplatin has an effect on CAR T cell proliferation. However, unfortunately, we cannot obtain enough cells for proliferation assays from the residual tumors following treatment.

Revision: Secretion has been changed to production within the text (page 24, line 566).

5. The authors have shown an enrichment of mouse macrophages showing M1 polarization in carboplatin-treated PDX tumors in immunodeficient mice. Do you expect the same effect on human macrophages? This aspect should be considered/discussed.

Response: To our knowledge, this is unknown and a completely novel result. The evidence from our animal studies suggests that the myeloid compartment plays a key role in the mechanism of action. Only time and clinical experience will tell whether this also occurs in humans, but it is of high relevance to the efficacy of this combination therapy, as well as other chemotherapy-immunotherapy combinations.

Revision: Statement relating to human macrophages added to discussion (page 32, line 759-762).

Minor comments:

1. Would be good to explain the treatment plan for nivolumab described in the Methods part in the section PDX experiment (page 10, line 232-234).

Revision: Nivolumab was used at 200 µg/dose, which is equivalent to 8 mg/kg for a 25 g mouse. This is a safe dose for human (PMID: 27879974) and a tolerable dosage for NSG mice based on our previous experiments for combination CAR-T therapies, as well as others' experience on nivolumab in PDX models (PMID: 31186248). Dose information has been included in the methods (page 11, lines 248-250).

2. In the Methods part (page 12, line 271) a bracket is missing.

Revision: Amended.

3. In Figure 1 legend: Please check whether tumors were considered positive with >10% or ≥10% membrane Le^Y-positive cells. (In the text is written "≥10% membrane Le^Y-positive cells".)

Revision: Text revised to >10% in the results (page 21, lines 475 and 480) and figure legend.

4. In Figure 1: Please define "NE" markers in the figure legend.

Revision: We have defined "NE" in the **Fig. 1** legend.

5. In Supplementary Figure 1a: Please define "Iso" in grey in the figure legend.

Revision: Amendment made defining "Iso" in **Fig. 1** legend.

6. In Supplementary Figure 1b: You have performed statistical analysis CAR-T versus media and IFN-γ versus CAR-T, correct? You should also include statistical analysis for IFN-γ versus media.

Revision: Statistical analysis for IFN-γ versus media is included in Supplementary Fig. 2c.

7. In Figure 3: Please change the labeling 3"E" into 3"e".

Revision: Amended.

8. In Figure 4: Please explain the abbreviation "DTX" in the figure legend.

Revision: Amendment made defining "DTX" in the **Fig. 4** legend.

9. In the Results part describing the CAR T cell activation and cytokine measurements you refer to Fig.2i-j (page 20, line 456-457). Should this be Fig.4i-j?

Revision: Amendment made on page 24, line 568. This figure is now to **Fig. 4h-i**.

Reviewer #3 (expertise in CAR T cells, T cell engineering):

Major comments:

1.1. Figure 5: Authors demonstrate that monotherapy with carboplatin alone results in an effective elimination of tumor cells 3 weeks after therapy (2.1% tumor cells), underlining that carboplatin-therapy alone is already very efficient in eliminating tumor cells. However, according to Figure 3D, carboplatin has only a minor effect on tumor size. How do authors explain these discrepancies? Are tumors measured in Figure 3D mainly consisting of TME rather than tumor cells? If so, which additional effects do Le γ CAR-T cells exert? Is there a cross-reactivity of the anti-Le γ binding domain of CARs with murine Le γ expressed e.g. on murine TME? Please provide in addition to Figure 5A, a diagram showing tumor volume changes in vehicle and carboplatin group over time.

Revisions:

- For greater clarity, the data for the vehicle and carboplatin alone treatment groups has been re-graphed on a linear axis in a new supplementary figure (**Supplementary Fig. 3a**).
- In **Supplementary Fig. 5a**, we have replotted the changes in tumour size across time for the vehicle and carboplatin treatment groups from Fig. 5a.
- To demonstrate consistency of the data in the two experiments, we have shown the percent change in tumor volume from carboplatin treatment each tumour from both experiments up to three weeks post treatment (**Supplementary Fig. 5b**). This shows no discrepancy between experiments as 97% (28/29) of grafts initially regressed across the two experiments.
- The text has been altered to refer to the sensitivity of PDX-287R to carboplatin treatment alone as follows (page 23, lines 521-524): *Whilst PDX-287R had an initial response to carboplatin treatment, this response was not sustained and tumors rapidly grew back, with tumors being harvested for tumor size at the same time as control grafts (Fig. 3d). In contrast, the addition of Le γ -CAR T cells resulted in a durable response to treatment, with no tumor regrowth for the duration of the experiment.*

1.2: Since most tumor cells were eliminated after 3 weeks, TME changes should be monitored at earlier time points. The CAR-T group should also be included.

Response: As described in response to reviewer 1 (minor comment 3), we have now included data from flow cytometry and RNA sequencing at one week post carboplatin treatment, which is the time of CAR T cell infusion. For PDX-287R, we found that carboplatin induced a decrease in the proportion of EpCAM⁺ tumor cells and an increase in tumor cell apoptosis within the grafts. RNA sequencing revealed a shift towards a pro-inflammatory phenotype in the tumor cells, as well as increased cGAS-STING signalling and increased expression of chemokines, including CXCR3 ligands for T cell chemotaxis. This suggests that carboplatin-induced cell death in PDX-287R tumor cells likely initiated a pro-inflammatory phenotype in prostate cancer PDXs, and could account for the increase T cell infiltration following carboplatin treatment.

At one week post treatment, there was also an increase in the proportion of CD45⁺ immune cells in PDX-287R grafts. Despite these proportional changes, RNA sequencing revealed that there was not a shift towards the M1 pro-inflammatory macrophage phenotype at this

stage, as we previously reported for the three-week time point. This suggests that this shift in immune cell phenotype occurs subsequently to tumor cell response.

Unfortunately, we could not include a CAR T cell group into these short-term experiments. However, we had already provided data showing that there was an increase in macrophage recruitment 48 hours after CAR T cell infusion in the carboplatin + CAR T cell group compared to the other treatment groups (**Fig. 4e-f**).

Revisions:

- Flow cytometry and RNA sequencing data for PDX-287R at the one-week time point has been included in a new figure, **Fig. 5**, with supplementary data in **Supplementary Fig. 4**. Corresponding text is on page 25 (lines 578-594), page 26 (lines 595-618) and page 27 (lines 619-620).

2: The authors have extensively studied carboplatin-induced changes within the TME. However, to obtain a complete picture of the underlying mechanisms of how carboplatin enhances the anti-tumor response of Le^Y CAR-T cells, the authors should also examine carboplatin-induced alterations on tumor cells: Do tumor cells change protein expression levels of Le^Y antigen upon carboplatin treatment? Do cancer cells alter the expression of co-stimulatory (e.g. 4-1BBL, Ox-40L, CD80, CD86) or co-inhibitory molecules (e.g. PD-L1, PD-L2, ...)? Do such carboplatin-induced changes on tumor cells, enhance the antitumor activity of CAR-T cells?

Response: Carboplatin did not alter the expression of Le^Y in PDX-287R following treatment *in vivo* based on immunohistochemistry staining. We also tested the response of PDX-287R organoids to carboplatin treatment *in vitro* over a 72 hour period, and found that carboplatin treatment did not change the protein expression of Le^Y using flow cytometry.

We looked for genes encoding co-stimulatory and co-inhibitory molecules in the RNA sequencing data at the one week time point; however, they were not detected.

Revision: The new data for Le^Y expression in treated PDX-287R grafts has been included in **Supplementary Fig. 2e-f**, with corresponding text on page 23 (line 529-534).

3. To analyze the effect of carboplatin on the TME, the authors chose an immunodeficient mouse model in which most immune cells are absent. Normally, TME studies are performed in syngeneic models or other immunocompetent mouse models. The authors should discuss the advantages and disadvantages of their model. How might the results observed in immunodeficient mice translate to humans?

Revision: New paragraph has been included in the discussion (page 33, lines 779-781; page 34, lines 782-792) as follows:

In this study, the action of carboplatin was assessed in immunosuppressed NSG mice, which lack T cells, B cells and natural killer cells. This was necessary to prevent xenogeneic rejection. However, an unappreciated benefit of this immunosuppressed model is that it allows the focus to be on the interaction between the tumor and human CAR T cells without a cognate effector immune response being induced through the murine lymphoid compartment. This lack of lymphoid cells has the advantage that we do not obscure the major shift we saw in the phenotypes of host myeloid cells, fibroblasts, ECM and endothelial cells in response to carboplatin. This is likely to be a critical facet of the mechanism of action, and highly relevant to the human setting. Of course, the immune system of prostate cancer patients does feature a fully functional lymphoid compartment, so it will be important that

our future mouse studies be extended into an immunocompetent syngeneic setting. We are currently designing experiments that meet this need.

Minor comments:

4. The authors should comment on the chosen dosing regimen and the applied concentrations of cisplatin, carboplatin and nivolumab in their combination studies with CAR-T cells. Are the applied concentrations clinically relevant/achievable. Were the drugs used similarly to clinical settings?

Response: Nivolumab was dosed at 200 µg/dose, which is equivalent to 8 mg/kg for a 25 g mouse. This is a safe dose for human (PMID: 27879974) and a tolerable dosage for NSG mice based on our previous experiments for combination CAR-T therapies, as well as others' experience on nivolumab in PDX models (PMID: 31186248).

Carboplatin was dosed at 50 mg/kg (single dose), which is equivalent to a human dose of 150 mg/m² (<https://www.fda.gov/regulatory-information/search-fda-guidance-documents/estimating-maximum-safe-starting-dose-initial-clinical-trials-therapeutics-adult-healthy-volunteers>). The dose for carboplatin used in the clinic is 400 mg/m², repeated once every three weeks. Docetaxel was dosed at 10 mg/kg (single dose), which is equivalent to a human dose of 30 mg/m². The dose for docetaxel used in the clinic is 75 mg/m², repeated once every three weeks. Therefore, the dose of these chemotherapeutic agents is lower than currently used in the clinic, and achievable in a clinical setting. We intentionally used a low dose to investigate the modulatory effect of each agent and its potential to synergise with CAR T cell therapy.

Revision: This information has been updated in the methods (page 11, lines 248-252).

5. Immunohistochemistry was used to analyze LeY expression in human prostate cancer samples. Based on which parameters were tumors classified as "low", "moderate" and "high"? What was the cut-off?

Revision: Low is classified as <10%, moderate 10-50% and high >50%. This has been included in the figure legend for Figure 1.

6. What was the transduction rate of CAR-T cells infused into the mice for treatment? If the transduction rate was not 100%, CD3 or CD8 staining cannot be used to measure infiltration of CAR-T cells. In this case, direct staining of CAR-T cells must be performed.

Revision: Additional methodology for the transduction of CAR T cells has been included (page 9, lines 210-212; page 10, lines 213-214). In the Le^Y-CAR construct, a CD34 epitope was embedded for purification. During the CAR-T cell production, the transduced T cells were purified using the EasySep™ Human CD34 Positive Selection Kit (STEMCELL Technologies) on day 4-5 post transduction. The purified cells were further expanded for infusion. The percentage of CAR⁺ cells in the final product was no less than 98%.

REVIEWERS' COMMENTS

Reviewer #1 (expert in prostate cancer, urology):

Regarding major comment #1, we still think it is better to include PDX models that show some level of response to docetaxel.
The rest of comments have been properly addressed.

Reviewer #2 (expert in CAR T cells, immunotherapy, prostate cancer):

All my concerns have been fully addressed in the revision. I have no further comments.

Reviewer #3 (expert in CAR T cells, T cell engineering):

After a thorough and substantial modification/amendment, the authors adequately addressed all major and minor comments I raised. Accordingly, the manuscript has been carefully revised. I therefore recommend that the manuscript by Porter et al. entitled "Tumor microenvironment modification by low-dose carboplatin augments CAR T cell efficacy in human prostate tumors" be accepted for publication in Nature Communications.

Response to reviewers:

Reviewer #1 (expert in prostate cancer, urology):

Regarding major comment #1, we still think it is better to include PDX models that show some level of response to docetaxel.

The rest of comments have been properly addressed.

Response: We have tested the response to docetaxel in 18 different PDXs in our collection, and none have shown a significant response *in vivo*. This is likely because our collection come from patients with high-risk, therapy resistant disease where the tumors are refractory to standard-of-care agents. Therefore, we are unable to test the effect of combination therapy with docetaxel in a docetaxel-sensitive tumor. However, we agree that there is evidence in the literature suggesting that docetaxel may also improve the efficacy of immunotherapy. To address this, we have included a paragraph in the discussion regarding the reports of docetaxel remodelling the immune microenvironment and enhancing checkpoint inhibitor-based immunotherapy of PSMA CAR T therapy in prostate cancer (page 20, lines 526-541).

Reviewer #2 (expert in CAR T cells, immunotherapy, prostate cancer):

All my concerns have been fully addressed in the revision. I have no further comments.

Reviewer #3 (expert in CAR T cells, T cell engineering):

After a thorough and substantial modification/amendment, the authors adequately addressed all major and minor comments I raised. Accordingly, the manuscript has been carefully revised. I therefore recommend that the manuscript by Porter et al. entitled "Tumor microenvironment modification by low-dose carboplatin augments CAR T cell efficacy in human prostate tumors" be accepted for publication in Nature Communications.